

**Is there evidence for a 4.2ka B.P. event in the northern North Atlantic region?**
Raymond S. Bradley[1] and Jostein Bakke[2]
[1]Department of Geosciences/Climate System Research Center, University of Massachusetts,
Amherst
[2]Department of Earth Science/Bjerknes Centre for Climate Research, University of Bergen
**Abstract**
We review paleoceanographic and paleoclimatic records from the northern North Atlantic to assess
the nature of climatic conditions at 4.2ka BP, which has been identified as a time of exceptional
climatic anomalies in many parts of the world. The northern North Atlantic region experienced
relatively warm conditions in the early Holocene (6-8ka B.P.) followed by a general decline in
temperatures after ~5ka B.P., which led to the onset of Neoglaciation. Although a few records do
show a distinct anomaly around 4.2ka B.P. (associated with a glacial advance), this is not
widespread and we interpret it as a local manifestation of the overall climatic deterioration that
characterizes the late Holocene.
**1. Introduction**
Detailed studies of two sediment cores in the North Atlantic (at ~65° and ~54°N) by Bond et al
(1997) revealed quasi-periodic variations in the percentage of hematite-stained grains and
Icelandic glass during the Holocene, which were interpreted as evidence for pulses of ice-rafting.
They argued that during these episodes, "*cool, ice-bearing surface waters shifted across more than*
*5° of latitude, each time penetrating well into the core of the North Atlantic Current*". One of the
8 Holocene episodes (later dubbed "Bond events") occurred at ~4.2ka calendar years B.P.
Subsequently, Bond et al. (2001) argued that these colder episodes were driven by changes in solar
insolation (cf. Wanner and Bütikofer, 2008; Wanner et al., 2011), notwithstanding the fact that
total solar irradiance did not vary by more than ±0.15% over this period (Vieira et al., 2011; Roth
and Joos, 2013; Wu et al. 2018). Other paleoceanographic studies have been unable to reproduce
the record of ice-rafting reported in Bond et al., (1997) (e.g. Andrews et al., 2014) yet the literature
is replete with studies that have tried to identify a signal linked to the timing of Bond events in
other paleoclimatic records from around the world (e.g. Fleitmann et al., 2003; Gupta et al., 2003;





Wang et al., 2005; Pèlachs et al., 2011). Here we review sedimentary records from the northern North Atlantic (north of 60°N) with a focus on evidence for an "event" around 4.2ka B.P. This region is of significance as it is the core region for ventilation of the North Atlantic which drives the Atlantic Meridional Overturning Circulation (AMOC), with global teleconnections through the conveyor belt system of ocean currents. We do not focus on records from Iceland as these have been reviewed separately by Geirsdóttir et al. (2018).

The North Atlantic has a very distinct pattern of sea surface temperatures, reflecting the ocean currents that traverse the region (Figure 1). Warm sub-tropical water enters the region from the southwest via the Gulf Stream (North Atlantic Current) and this transfers heat to sub-polar latitudes north of Scandinavia by way of the Norwegian Atlantic and West Spitsbergen currents, as well as around the western and northwestern coast of Iceland via the Irminger current. In contrast, cold polar water exits the Arctic Ocean via the East Greenland current, which extends to around the southern tip of Greenland. The region between these water masses is where deepwater formation occurs, driving the large-scale Atlantic Meridional Overturning Circulation (AMOC). On the timescale of the Holocene, there have been significant changes in the characteristics and position of these major oceanographic features, as recorded by various paleoceanographic proxies.

## 2. Paleoceanographic evidence

First, we consider a transect of sediment cores that are aligned along the axis of the main influx of Atlantic water entering the North Atlantic, from north of the UK to west of Svalbard (Figure 1) Sea-surface temperatures have been tracked using alkenones and diatoms, which reflect conditions in the photosynthetic mixed layer of the ocean surface, and by the relative abundance of *Neogloboquadrina pachyderma* (s), which is diagnostic of cold polar water (Figure 2). All studies reveal higher SSTs in the early Holocene, with the largest anomalies (relative to today) at high latitudes (that is, there was strong polar amplification of the warming) (Andersson et al., 2010). This early Holocene warming was a consequence of orbital forcing: June/July insolation was ~10% higher than today at the start of the Holocene in the northern parts of the region, but the peak warming was delayed due to the influence of the decaying Laurentide and Scandinavian Ice Sheets and associated icebergs and freshwater (Renssen et al., 2009, 2012; Zhang et al., 2016). Consequently, maximum temperatures were a few thousand years later than the peak insolation, punctuated by a short-lived cooling event around 8.2ka B.P. associated with the final major



freshwater discharge event of the Laurentide Ice Sheet (Barber et al., 1999; Rohling and Pälike,
2005). Thereafter, as insolation declined so sea surface temperatures declined steadily, or by some
estimates, in a more step-like manner (e.g. Calvo et al., 2002; Risebrobakken et al., 2010).  For
example, Birks and Koç (2002), Andersen et al. (2004) and Berner et al. (2011) all found that
August SSTs at 67°N (core MD95-2011) were 4-5°C warmer than today from ~9000-6500 years
B.P., then steadily declined. These analyses were based on diatoms, but similar results (albeit with
a smaller change in temperature, ~2.5°C, perhaps reflecting a different seasonal bias) were
obtained in a study of alkenones from the same core (Calvo et al., 2002). Studies further north,
paint a similar picture (Sarnthein et al., 2003; Risebrobakken et al., 2003, 2010; Werner et al.,
2014). This pattern of maximum SSTs in the first half of the Holocene and cooling thereafter is
seen throughout the eastern North Atlantic, in all proxies that are indicative of conditions in the
photic zone (Rimbu et al., 2003; Leduc et al., 2010; Sejrup et al., 2016). The timing of the onset
of cooling varies, but in all cases cooling was well underway by~5.5ka B.P., in what some refer to
as a "transition period" that subsequently led to much cooler conditions in the late Holocene (after
3.5ka B.P.) (e.g. Aagaard-Sorensen et al., 2014; Andersen et al., 2004; Leduc et al., 2010; Sejrup
et al., 2016). Although there were short-lived cooling episodes superimposed on the overall first
order pattern of temperature change (e.g. Werner et al., 2014), there is no evidence for quasi-
periodic cooling episodes disrupting the northward flux of Atlantic water, as described by Bond et
al (1997).  Proxies of sub-surface conditions (below the mixed layer) – Mg/Ca ratios and oxygen
isotopes in forams, as well as foram assemblage changes – generally do not show the same pattern
of pan-Holocene cooling as the SST proxies, often indicating slight warming through the Holocene
(e.g. Andersson et al., 2010; Sejrup et al., 2011).  But these records also do not show a pattern of
quasi-periodic cooling events. Could this be because of low resolution in sampling, or poor
chronologies?  This seems very unlikely as many of these records are from high-deposition rate
sites, providing high resolution records that are generally well-dated (e.g. Berner et al., 2011).
Indeed, one exceptionally well-dated, high resolution sediment core from the Storegga Slide region
(90 AMS [14]C dates over 8000 calendar years) provides oxygen isotope data on planktonic forams
at a resolution of ±20 years within the core of the Norwegian Atlantic Current at ~64°N. This
clearly shows multi-decadal to century-scale variability throughout the last 8000 years, but none
of the cold water flux episodes that one would expect to see, based on the work of Bond et al.
(1997).  We therefore conclude that there is no signal of a 4.2ka B.P. event in paleoceanographic



proxies from regions influenced by the flux of warm water from the sub-tropical Atlantic into the
Nordic Seas. Cooling had set in more than a millennium earlier in this region.

Next, we consider studies in the western part of the North Atlantic, north of Iceland on the

Icelandic Shelf, and further to the east, near Denmark Strait. Here, many studies have examined,
*inter alia*, foraminiferal assemblages, coccoliths, dinoflagellate cysts and sea-ice biomarkers and
ice-rafted debris (IRD) reflecting transport of material in the cold East Greenland Current (e.g.
Andrews et al., 1997; Jennings et al., 2002; Giraudeau et al., 2004; Solignac et al., 2006; Sicre et
al., 2008; Justwan et al., 2008; Perner et al., 2015; Moossen et al., 2015; Cabedo-Sanz et al., 2016;
Kolling et al., 2017). In this region, warmest conditions occurred around 6.0±1.5ka B.P. (the
timing depending on location); these conditions were associated with minimal input of IRD,
reflecting the recession of tidewater glaciers onto land along the eastern coast of Greenland, and a
weak East Greenland Current, with minimal stratification of the water column at that time as the
flux of warmer, more saline Irminger Current water increased (Justwan et al., 2008; Jennings et
al., 2011; Werner et al., 2014; Telesinski et al., 2014; Perner et al., 2016). Conditions began to
change by ~5.0±0.5ka B.P. (the timing varying geographically) when cold water diatoms and
forams, sea-ice (as tracked by the biomarker index, $IP_{25}$) and IRD started to increase, and the water
column became more stratified as the East Greenland Current strengthened (Moros et al., 2006;
Telesinski et al., 2014; Perner et al., 2016; Kristiansdottir et al., 2017). These changes correspond
to the re-advance of glaciers in East Greenland, part of the much more widespread onset of
neoglaciation that is well-documented in many regions around the North Atlantic (Solomina et al.,
2015).Warmer conditions (strengthened Irminger Current) developed over the past 2000 years, but
this period is also characterized by a series of minor fluctuations in the extent of ice in the region,
with much colder conditions after ~1.0ka B.P. when the coldest conditions of the last 8000 years
occurred, with abundant IRD and sea-ice in Denmark Strait and off the north coast of Iceland
(Bendle and Rosell-Mele, 2007; Andresen et al., 2013; Cabedo-Sanz at al 2016; Kolling et al.,
2017). None of these records show evidence of an unusual anomaly at 4.2ka B.P.; rather, the
overall cooling of the late Holocene began 500-1000 years earlier (cf. Orme et al., 2018). Similar
variability is also seen further south and southwest of Iceland, at ~59°N (Farmer et al., 2008; Moros
et al., 2012; Orme et al., 2018) though there is evidence from dinocysts for an anomaly in the
seasonality of SSTs at ~4.5ka B.P., perhaps related to a westward shift in the Sub-Polar Gyre,
allowing warmer Atlantic water to influence the site (van Nieuwenhove et al., 2018).



This review of paleoceanographic studies extending from southern Greenland to Fram
Strait, and from western Svalbard and the southern Barents Sea southward to 60°N, provides no
evidence for a significant change in major oceanographic conditions that could be linked to the
4.2ka B.P. climate anomaly seen elsewhere.  Rather, the evidence points to a more gradual change
that was well under way by ~5ka B.P., from the relatively warm conditions of the early Holocene
(driven by precessional forcing) to much colder conditions that have characterized the last 3
millennia.

**3.  Terrestrial records from around the North Atlantic**
**3.1 Eastern Greenland and the Greenland Ice sheet**
Lake sediment records from sites along the coast of eastern Greenland provide a record of
Holocene environmental conditions that generally reinforce the paleoceanographic evidence
discussed earlier.  A "Holocene Thermal Maximum" (characterized *inter alia* by longer ice-free
conditions, higher levels of lacustrine productivity, increased evaporation, more tundra vegetation
and higher levels of terrestrial plant material transferred to lakes) is clearly seen from ~8ka B.P.
(or earlier) to ~5.0±0.5ka B.P (e.g. Kaplan et al., 2002; Andresen et al., 2004; Schmidt et al., 2011;
Balascio et al., 2013; Wagner and Bennike, 2015; Axford et al., 2017; van der Bilt et al., 2018a).
Thereafter, conditions became colder, often with a decline in vegetation cover, an increase in the
flux of coarse-grained sediments, and a shift in the types of chironomids and diatoms present,
towards species that thrive in cooler conditions.  At the same time, in glacierized watersheds, the
growth of glaciers led to an increase in the flux of minerogenic material which is a diagnostic
signal of the onset of late Holocene neoglaciation across the region.  In Kulusuk Lake (65°N) this
change occurred at ~4.2ka B.P., when there was an abrupt increase in clastic sediments from
glaciers that had probably disappeared during the mid-Holocene warm period (Balascio et al.,
2015). A similar transition is seen in sediments from nearby Ymer Lake, where a higher frequency
of avalanches and a longer season with ice-cover is thought to have favored the transfer of coarser
material into the lake after ~4ka B.P.  At another site in the same region, the Holocene thermal
maximum was identified (via the evaporative enrichment of δD in leaf wax n-alkanes) from 8.4 to
4.1ka B.P, followed by a decrease in evaporation as the open water season became shorter.  At the
same time, there was an increase in the flux of clastic sediments and terrestrial organic material
into the lake as river runoff increased (Balascio et al., 2013).  In all of these studies, it is clear that





there was a fairly rapid transition from warm mid-Holocene conditions to the colder, wetter late
Holocene that encompassed the 4.2ka B.P. interval of interest.  In some cases, there is evidence
for a short-lived "event" at around that time (e.g., at Kulusuk Lake; Balascio et al., 2015) but this
appears to be simply part of the overall deterioration in climate that led to ice growth across the
region.  There is currently no evidence for a more widespread glacial advance at 4.2ka B.P. Given
that cooling was persistent over the last 5000 years, and the elevational threshold for glacierization
is close to mountain tops across the region (declining in elevation poleward), it is understandable
that different locations would have experienced the onset of neoglaciation at, different times.
However, as the ELA continued to lower over the last 3-4 millennia, glaciers that had greatly
diminished in size, or disappeared entirely, during the Holocene Thermal Maximum were
eventually regenerated, with the exact timing varying across the region. In the case of Kulusuk
Lake, it seems reasonable to conclude that the steady decline in temperatures and the specific
hypsography of that basin led to a positive mass balance, with early ice growth and associated
sediment input to the lake around 4.2ka B.P.

Ice cores from Greenland provide records of past climate variations from oxygen isotopes,

glaciochemistry and physical characteristics, which are broadly consistent with those from coastal
lake sediments. Alley and Anandakrishnan (1995) examined evidence for summer melting in the
GISP2 ice core, as recorded by changes in the physical properties of the ice.  Their analysis was at
a relatively low resolution, but they showed maximum Holocene summer temperatures from
~7.5ka B.P., followed by a two-step transition to colder conditions, from ~6.5 to 5.5ka B.P., and
~4.5 to 4ka B.P., with persistently low summer temperatures (minimal melting) thereafter.  After
adjusting for ice thickness changes, Vinther et al. (2009) also showed that there was an overall
decline in temperature at the Summit of the Greenland Ice Sheet (73°N, 3210 masl) over the last
~9,000 years (interpreted from changes in $\delta^{18}O$ in the GISP2 ice core) with the warmest 20 year
period ~7970 years b2k, and the coldest ~300 years b2k. These two periods differed in mean
temperature by ~4.9°C (though it is unclear if this was mean annual temperature as small changes
in the seasonality of snowfall on the ice sheet could have drastically changed the apparent
temperature over time).  Superimposed on the long-term temperature decline there were
multidecadal anomalies on the order of ±1°C.  One of the largest of the negative anomalies after
the well-known 8.2ka B.P. event began ~4400 b2k and reached a minimum at 4340 b2k, but by
4200 b2k, temperatures had sharply increased (Figure 3). The 8.2ka B.P. cooling episode was the



result of freshwater flooding of the North Atlantic as the Laurentide Ice Sheet collapsed, so a
different explanation is required to explain the later anomaly, and it seems plausible that this short-
lived cooling event was a consequence of the massive eruption of Hekla (in Iceland) at ~4.2ka B.P.
(recognizing that there are dating uncertainties in both the ice core and tephrochronological
databases). Kobashi et al. (2017) also reconstructed mean annual temperature, derived from the
differential diffusion of argon and nitrogen isotopes in firn prior to its densification into ice. This
provided a temperature record which is similar to that of Vinther et al (2009) but with more multi-
decadal to century-scale variability. Although it also shows a negative temperature anomaly
around 4.5ka B.P., this is well within the normal variability of the record; a more significant
temperature decline is seen somewhat later, centered on ~3.5ka B.P. In summary, there is no
compelling evidence for a distinct climatic anomaly at 4.2ka B.P. in ice cores from Greenland.

**3.2 Svalbard**

Lake sediment records from Svalbard record changes in climate at the northernmost limit of North
Atlantic water (the West Spitsbergen Current). All studies describe a warm early Holocene phase
when many of the glaciers seen today were small or absent. On Amsterdamoya, at the northwestern
edge of Svalbard, warm and dry conditions spanned the interval from 7.7 to 5ka B.P., and nearby
glaciers were small or absent by 8.4ka B.P., only re-forming in the late Holocene (Gjerde et al.,
2018; de Wet et al., 2018). To the southwest, on the Mitrahalvoya Peninsula, there is also evidence
that glaciers reached their minimum size by the mid-Holocene, but subsequently re-formed or re-
advanced. Karlbreen began to grow around 3.5ka B.P. (Røthe et al., 2015) but in the neighboring
watershed of Hajeren an abrupt increase in minerogenic sediments at 4.25 ka B.P. registered the
onset of neoglaciation in the basin (van der Bilt et al., 2015). Paleotemperature estimates (from
alkenones) in the same record indicate this advance was triggered by an abrupt drop in temperature
at that time; thereafter, temperatures remained low (van der Bilt et al., 2018b). Other records from
the region indicate that the first neoglacial advances of glaciers occurred around 4.6ka B.P. (e.g.
Svendsen and Mangerud, 1997; Reusche et al., 2014).

**3.3 Scandinavia**

As most glaciers in Scandinavia had their largest areal extent during the "Little Ice Age" (~A.D.
1400-1850), information about past glaciers in Norway during the late Holocene is based on



reconstructions from indirect evidence, mainly sediments deposited in distal glacier-fed lakes (e.g.
Nesje 2009, Bakke et al., 2010; 2013). After several large glacier advances in the earliest Holocene,
the climate was in general warm during the early Holocene (8.5-6.5ka B.P.) and most glaciers
melted away (Nesje 2009) (Figure 4). Around 6 ka B.P. glaciers start to re-grow mainly as a
function of decreasing summer insolation over the Northern Hemisphere (Wanner et al. 2008). The
regrowth of glaciers shows a gradual increase in glacier size interrupted by smaller glacier
advances (Bakke et al, 2010, 2013; Vasskog et al., 2012). Along a coastal south-north transect in
Scandinavia different locations have experienced the onset of neoglaciation at different times,
mainly as a function of altitude (cf. Geirsdóttir et al., 2018). Around 2ka B.P. many glaciers
reached present day size with a maximum glacier extent during the Little Ice Age (Nesje 2009). A
review of more than 20 papers shows that none of them indicate any abrupt anomalous change in
glacier extent connected to a perturbation of climate around 4.2 ka. (Bakke et al., 2005a; 2005b;
2008; 2010; 2013; Dahl and Nesje; 1992; 1994; 1996; Lauritzen 1996; Snowball and Sandgren,
1996; Seierstad et al., 2002; Lie et al., 2004; Nesje et al. 2009; Vasskog et al., 2011; 2012 Støren
et al., 2008; Wittmeier et al., 2015; Shakesby et al., 2007; Kvisvik et al., 2015, Gjerde et al., 2016).
Investigating this further, we examined other terrestrial evidence mainly pollen, macrofossil and
diatom records derived from lake sediments (e.g. Bjune et al., 2005; Velle et al., 2005). They have
a time resolution somewhat lower than the glacier reconstructions (typical 500 yr spacing) but they
all reflect the general decrease in summer insolation over the northern hemisphere and no abrupt
transition close to 4.2ka B.P. (Bjune, 2005; Bjune et al., 2004, 2006; Velle et al., 2005). The only
terrestrial evidence from Scandinavia that shows a clear anomaly at 4.2ka B.P. is a speleothem
record of $\delta^{18}$O from Northern Norway (Lauritzen and Lundberg 1999) where higher temperatures
are recorded, peaking at 4.2ka, before a rapid decrease to much colder temperatures at ~3.7ka B.P.

**4. Conclusions**
A review of paleoceanographic and terrestrial paleoclimatic data from around the northern North
Atlantic reveals no compelling evidence for a significant climatic anomaly at ~4.2ka B.P. In
particular, there is no supporting evidence for "*cool, ice-bearing surface waters…penetrating well*
*into the core of the North Atlantic Current*" at that time, as described by Bond et al., (2001). The
region experienced relatively warm conditions in the early Holocene (6-8ka B.P.) followed by a
general decline in temperatures after ~5ka B.P., signaling the onset of Neoglaciation. Although a





few records do show a distinct anomaly around 4.2ka B.P. (associated with a glacial advance), this
is not widespread and we interpret it as a local signal of the overall climatic deterioration that
characterized the late Holocene.

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

Holocene warming in the extratropical Northern Hemisphere, *Climate of the Past*, 12, 1119-
1135, **2016**.




















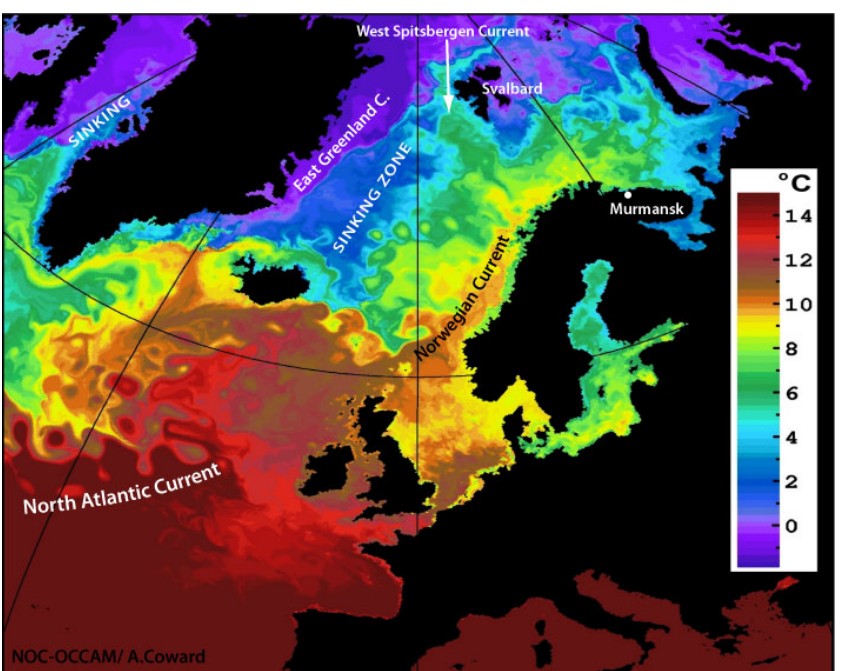





**Figure 1**. Major ocean currents in the North Atlantic and associated sea surface temperatures.
(Source: NOC/UK Met-Office OSTIA data; map from http://www.seos-project.eu)



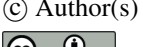



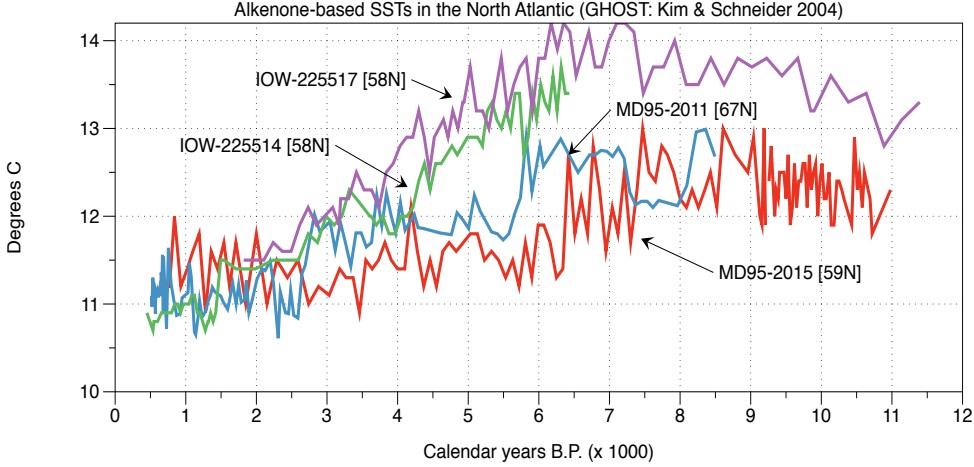



**Figure 2.** Holocene August SSTs at various locations in the northern North Atlantic (Anderson et
al., 2004) and alkenone-based SSTS from sediment cores along a N-S transect in the North
Atlantic Current-Norwegian Current system (cf. Figure 1).















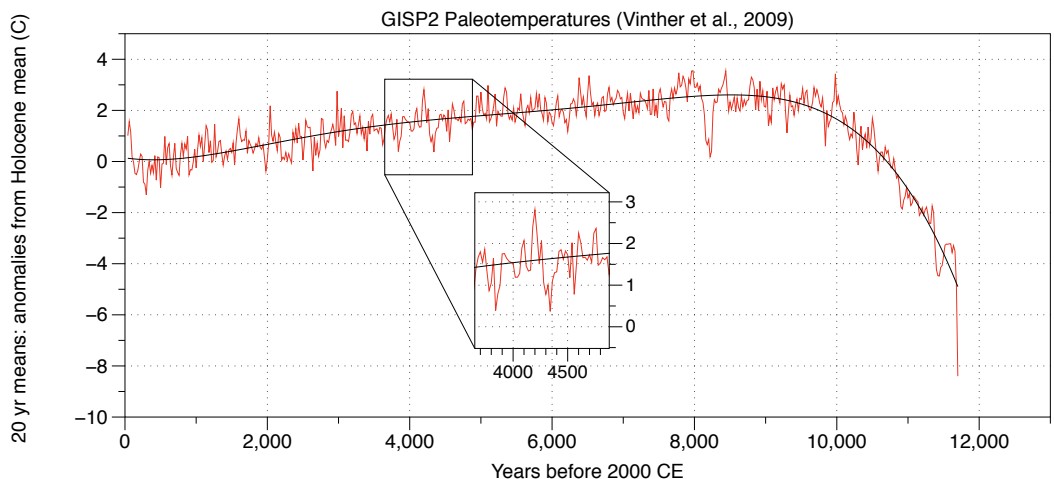



**Figure 3.** Oxygen isotope anomalies ($\delta^{18}$O) relative to the Holocene average. Timescale is in years
b2k (before A.D. 2000). The interval around 4.2ka BP is enlarged in the box (Data source:
Vinther et al., 2009).














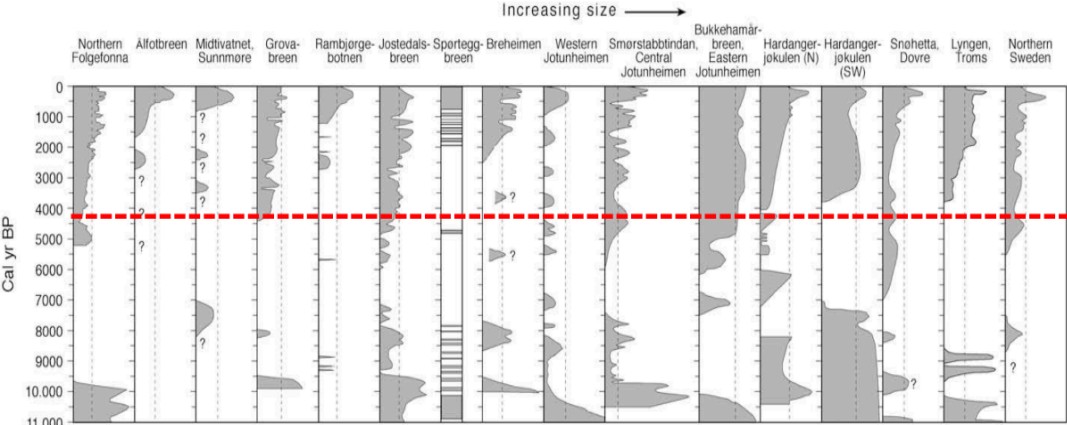




**Figure 4.** Summary of glacier extent in various regions of Scandinavia during the Holocene. 4.2ka

B.P. is highlighted by the red dashed line (after Nesje, 2009).

