# Peer review of "Is there evidence for a 4.2ka B.P. event in the northern North Atlantic region?"

_Climate of the Past, 2018_

## Short Comment (SC1) · 29 Jan 2019

The caption of Figure 2 says: Holocene August SSTs at various locations in the northern North Atlantic (Anderson et al., 2004) and alkenone-based SSTS from sediment cores along a N-S transect in the North Atlantic Current-Norwegian Current system. I might be wrong here, but the records seem to be all alkenone records, as implied by the graph title. I don't think there are any data from Andersen (not Anderson) et al., 2004, in the graph. Andersen et al., is a diatom paper. Yes, there is a diatom record from MD95-2011, but the data presented for this core looks like the alkenone data by Calvo et al., 2002. Another thing, perhaps core locations could be added to Figure 1?
* * *

---

## Referee Comment (RC1) · Anonymous Referee #1 · 30 Jan 2019

Summary and major suggestions

This manuscript addresses an important and interesting central question – whether there is evidence for a prominent "4.2 ka event" in paleoceanographic and terrestrial paleoclimate records from the northern North Atlantic regions directly affected by the North Atlantic Current and East Greenland Current (excluding Iceland, which is covered in a parallel submission by other authors). This paper can make a valuable contribution, but it will benefit from some major additions:

1. It should be made clear exactly what records are considered in this study, and (if they are a subset of published records) why these records were chosen. For example, section 2 begins with "we consider a transect of sediment cores," but what sediment cores are referred to here, and why are these cores in particular singled out? Is the

"transect" a list of all available records the authors could find from the region? Or a subset of published records selected for some specific reason?

Choice of sites used in all parts of this study should be clearly explained (and justified if the study is assessing a subset of published records), preferably (for clarity) in a short "Methods" section (which is currently lacking from the paper).

Site locations should be added to Figure 1. Optionally, a table listing all sites, locations, proxies, and original publications would be useful.

2. In my opinion, all of the datasets described here should be shown in the paper's figures. (As an example, Fig. 4 is a nice way to visualize a large number of glacier reconstructions; although it is unclear whether that figure is updated with records obtained since 2009.)

It would also be useful to include a statistical summary/ analysis of the data, as an objective test for a 4.2 ka event.

Very few of the data used to support the paper's main conclusion are shown, and there is no particular methodology of site selection or analysis described. Thus, with the information provided, it is impossible for readers to evaluate or appreciate the evidence. I personally trust the authors' expert knowledge of records from the region – but that alone is not a strong enough basis for their conclusions to be published in COP.

3. The goal of this study is to assess whether a 4.2 ka event is a coherent feature of Holocene climate in the study region. Yet most of the text actually summarizes multi-millennial climate trends through the Holocene, and only relatively short sections of text discuss/evaluate evidence for 4.2 ka events in various regions. In the first part of section 2, for example, lines 55-84 summarize multi-millennial Holocene trends, then lines 85-94 assess higher-frequency events and conclude there is no 4.2 ka event. Same for lines 96-118 (which review multi-millennial trends) vs. lines 119-124 (which evaluate evidence for/against a 4.2 ka event). Given the goal of this study, I would

expect to see relatively more extensive discussion of high-frequency events, and the evidence for/against a prominent, coherent 4.2 ka event. If statistical analyses are added as suggested above, that will help flesh out the discussion.

Minor comments

What convention is used to subdivide early from middle from late Holocene? e.g., the Abstract refers to the period 8-6 ka BP as part of the early Holocene, though in many subdivisions this would be considered part of the middle Holocene.

Depending on the desired geographic scope of this analysis, additional terrestrial records that could be included from east Greenland include Levy et al 2013 QSR (Bregne ice cap) and Lowell et al. 2013 QSR (Istorvet ice cap). Neither is ideal to capture subtle climate changes ∼4.2 ka, but these records may help make the authors' point about idiosyncratic glaciation thresholds for individual glaciers/catchments.

Line 19: It appears this paper is to be part of a special issue focused on the 4.2 ka event, so I understand why there is not an introduction to the broader (global/hemispheric) significance/question of the 4.2 ka event. Nonetheless, it would be helpful for future readers (who may read this paper on its own) if the Introduction began with at least a couple of sentences of background on the 4.2 ka event more broadly, rather than beginning by discussing Bond events, to clarify why testing for climate shifts at 4.2 ka (vs. shifts correlating with other Bond events) is of particular interest.

Line 37: The Geirsdottir review of Icelandic records appears to be published (or nearly). These findings from Iceland seem highly relevant to the current study and should be briefly summarized and discussed somewhere (Section 3.3?).

Line 75: "in all cases" SST cooling specifically? Meaning all published studies? Or cores in Fig. 2?

163: Omit extra comma after "at"

Line 181: Why compare the coldest and warmest 20-year periods of the Vinther record, rather than stick to describing millennial-scale changes? Comparing the warmest and coldest millennia would seem more consistent with the rest of the discussion of multi-millennial scale changes. As currently written (e.g. followed immediately by the phrase "Superimposed on the long-term temperature decline...") 4.9 C comes across as an estimate of the long-term, multi-millennial temperature change inferred from the Vinther reconstruction (for which it would be a major overestimate).

Line 249: The concluding sentence states that "Although a few records do show a distinct anomaly around 4.2 ka BP (associated with a glacial advance) . . . we interpret it as a local signal of overall climatic deterioration that characterized the late Holocene." Is this consistently the onset of (subsequently more-or-less continuous) glaciation, as opposed to a transient, short-term glacial advance? If the former, which is what I gather from the previous discussion, then stating that more clearly here would strengthen this final assertion of the paper.

Fig 1. Locations of study sites mentioned in the text should be shown in this figure.

Fig. 2. Why are data plotted only from these sites?

Fig 3. It would be useful to see Kobashi et al. 2017 also plotted, given the different sub-millennial variability in that record.

---

## Referee Comment (RC2) · Harvey Weiss (Referee) · 26 Mar 2019

Harvey Weiss (Referee)

harvey.weiss@yale.edu

The 4.2 ka BP event in the northern North Atlantic.

Harvey Weiss1,

1 School of Forestry and Environmental Studies, Yale University

In the northern North Atlantic, the 4.2 ka BP event is evident in lake, bog, marine, glacial, speleothem and tree ring cores with extensive, coherent, and high resolution proxy data for abrupt century-scale alterations of temperature and precipitation. These records extend across the northern North Atlantic, 1900 kms northeast to southwest, from Spitzbergen, Svalbard to Agassiz Ice Cap, Ellesmere Island, including Sweden, Norway, Denmark, Faroe Islands, Iceland and adjacent seas, and Greenland. Adjacent

region, high resolution proxy data in Europe and North America provide synchronous and similar records. The proposed article by Bradley and Bakke (cp-2018-162, in review), however, ignores the relevant data from Svalbard, Sweden, Norway, Denmark, Faroe Islands, Iceland, Nordic Seas, Greenland and Ellesmere Island.

In Figure 1, a) - b) are Greenland Ice Sheet Total mass balance and ice volume experiments 5 and 6 from Nielsen et al, 2017 that present an abrupt ca 200 year warming event beginning at ca 4.3 ka BP. This melt spike is synchronous with c), the modelled 4 degree SST cooling spike in the Northwest Atlantic, ca. 4.3-4.1 ka BP (Klus et al, 2017) and with the abrupt NGRIP ca. 5 degree K warm spike ca. 4.5-3.9 ka BP (Gkinis et al, 2014). In d), the Agassiz, Ellesmere Island and Renland, Greenland ice core temperature spike is 3-stage, beginning at 4290 BP, reaching its apogee at 4150 BP, returning to baseline at 3990 BP, and descending to pre-event levels at 3790 BP (Vinther et al., 2009). The sudden GISP2, Greenland temperature spike (Kobashi et al, 2017), although less well-defined, conforms to this event. These six key North Atlantic high resolution and modeled data are summarized in e), which presents the remarkable congruence of the Lake Hajeren, Svalbard, sediment core and the Agassiz, Ellesmere Island ice core. The Lake Hajeren neo-glaciation spike, recognized in minerogenic / glacigenic indicators TDBD (dry bulk density) and Ti/Loss on Ignition z-scores from core HAP0212, extends from ca. 4250 BP to ca. 4100/4050 BP, a calibrated radiocarbon interpolation across two hundred years (van der Bilt et al, 2015). The synchronous Agassiz ice core melt spike extends from ca. 4250 – 3950 BP, with an error of ca. 20 years (Fisher et al, 2012; Lecavalier et al., 2017).

In summary, the Lake Hajeren, Spitsbergen, Svalbard cold glaciation event was synchronous with the Ellesmere Island Agassiz ice core warm melt event 1900 kms distant across the span of Island and adjacent seas and Greenland. The same relationship obtains with the NGRIP warm event (Gkinis et al, 2014) and the modeled Northwest Atlantic Sea Surface Temperature event (Klus et al, 2018): in the northern North Atlantic, cold lake and sea events were synchronous with warm, elevation-corrected, glacier

events that extend as far west as Mount Logan, Yukon (Fisher et al, 20128). This curious, highly resolved, 4.2 ka BP event situation has not been discussed previously and there exist neither proximate nor ultimate explanations for it.

Svalbard

The congeries of five relevant lake sediment studies on Svalbard utilizes a variety of paleoclimate proxies, of which the most sensitive display a clear 4.2 ka BP abrupt cooling event. Chironimid analyses from Lake Svartvatnet (Luoto et al., 2017) and a leaf wax study at Lake Hakluytvatnet (Balascio et al., 2018) show no evidence for 4.2 ka BP climate events. In Lake Hakluytvatnet, one study indicates a spike of "increased run off intensity" representing significant sea ice alterations, and a spike in XRF Si/Ti suggests decreased lake productivity "reflecting milder and wetter (i.e., more maritime conditions)" between 4200 and 3700 BP (Gjerde et al, 2018); these are, however, only indirect climate proxies. Definitively, the alkenone paleothermometry at both Lake Hakluyvatnet and Lake Hajeren (van der Bilt et al 2018) are supported significantly by the minerogenic/glacigenic indicators at Lake Hajeren (van der Bilt et al., 2015). A two-step Holocene cooling is defined, "with transitions between ∼7.8-7 ka cal. BP and after ∼4.4-4.3 ka cal. BP". The abrupt transition after 4.4-4.3 cal ka BP is "best captured by a 2 degree C temperature decrease between ∼4.4-4.3 and 4.2 cal ka BP. . . with short-lived glacier re-growth in the catchment around 4.25 ka cal. BP" that extended to ca. 4.05 cal. BP (van der Bilt et al 2018).

For the Svalbard 4.2 ka BP event proxies, Bradley and Bakke cite van der Bilt et al., 2015 Lake Hajeren, whereas there are five lake studies from Svalbard. For Lake Hakluytvatnet, Gjerde et al., 2018 is misrepresented, while van der Bilt et al., 2018 for Lake Hajeren and Lake Hakluytvatnet is not mentioned.

Sweden

Adjacent regions' paleoclimate proxies display similar cold and wet 4.2 ka BP events. Four such records are in Sweden. At Lake Igelsjön, southern Sweden, a lake sediment

core revealed "marked and coherent depletions in 18O and 13C at ca 4000 cal BP" (Hammarlund et al, 2003). At Lake Trehörningen, in southwest Sweden, the lake sediment pollen analysis indicates that the warm temperate tree taxa, Tilia (Linden) and Ulmus (Elm), decline beginning at 4K cal yr BP, due to a "a predominantly climatic retreat" (Antonsson and Seppa 2007). In central Sweden, moisture sensitive Scots pines (Pinus sylvestris L.), bog-preserved logs sampled from small lakes, define annual resolution lower lake-levels 2400–2200 BC and 2100–1800 BC (Gunnarsson 2008). Similarly, at Åbuamossen, southern Sweden, a 1561-year tree-ring width chronology was developed from 159 Scots pines. The earliest of three main wet-shifts here is precisely dated 2150-2100 BC, and likely "related to the to the stepwise Mid- to Late Holocene climate transition, during which the condition changed from relatively warm and dry towards cold and moist in the northern hemisphere "(Edvardsson 2016). Synchronous dying off phases during increasingly wet conditions are recorded at Venner Moor, Germany (Eckstein et al., 2010).

None of these Swedish 4.2 ka BP event proxies are mentioned by Bradley and Bakke.

Norway

Four proxies record the 4.2 ka BP event in Norway. At Søylegrotta, northern Norway, calibration of the isotope record from speleothem sample SG93 defines the 3-stage 4.2 ka BP cooling event that began at 4220 BP to 4035 BP with an abrupt temperature increase from 2.8 deg C to 4.6 deg C, i.e., 1.8 deg C in 185 years. This was followed 4035-3730 BP by an abrupt temperature decrease from 4.6 deg C to 1.6 deg C, i.e., 3 deg C cooling across 305 years, and a third stage temperature rise to 3 deg C by 3600 BP (Lauritzen and Lundberg 1999). The second proxy event, also in northern Norway, is a distinct glacier advance reconstructed between 4420 ± 45 and 4300 ± 40 cal. yr BP at Leirdalsbreen that "is suggested to indicate the start of the Neoglaciation at Høgtuva," (Jansen, et al, 2016). The third Norway proxy is the synchronous glacial advance observed at Austre Okstindbreen, with a dry bulk density spike at 4.2 ka BP, "an event arguably global in scope" (Bakke et al., 2010). The fourth proxy comprises the two

lakes at Lofoten Islands that show abrupt transitions to wetter conditions at 4.3 ka BP, as indicated by radiocarbon dated macrofossils, dry bulk density, and sedimentation rates (Balascio and Bradley 2012).

Bradley and Bakke do not mention the Leidalsbreen glacier advance, the Austre Okstindsbreen glacial advance, nor the Lotoften Islands abrupt transitions to wetter conditions. They claim, ll. 228, for Scandinavia, "A review of more than 20 papers shows that none of them indicate any abrupt anomalous change in glacier extent connected to a perturbation of climate around 4.2 ka." Their examination of the terrestrial evidence concludes, ll. 236, "they all reflect the general decrease in summer insolation over the northern hemisphere and no abrupt transition close to 4.2ka B.P."

Denmark

Synchronous with the Swedish and Norwegian proxy data, the recently retrieved sedimentary sequence at Filsø, a coastal wetland in western Denmark, indicates an intense, large scale aeolian sand influx at unit III: "a sharp transition to a 15 cm-thick bed of dune-sand which was dated to 4100 ± 200 B.P. and undoubtedly corresponds to the period of enhanced aeolian activity and intense dune movement identified for the same period along the entire western coast of Denmark" (Goslin et al 2018). This Filsø storm period, ca. 4400-3800 BP, may be related to the synchronous northward shift of the Azores Front (Repschläger, et al., 2017).

Bradley and Bakke do not mention the Filsø sediment core.

Faroe Islands

There are three reports of the 4.2 ka BP event from the Faroe Islands. Sediment cores at Streymoy's Lake Starvatn and Sandoy's Lake Lykkjuvøtn have a Zone 4 that begins abruptly at 4200 cal yr BP, according to high resolution radiocarbon dating, with decreases in biogenic silica and increases in sand grains flux, that indicate increase in lake ice and windiness (Andresen et al 2006). Second, a piston core from the Faroe

east shelf, previously studied with radiocarbon dates and sedimentation rates, indicates the lowest SST from 4000 BP based on the distribution of planktic and benthic foraminifera, accumulation rates, $\delta$18O values and calculated temperatures and salinities (Rassmussen, et al., 2010). Third, studies of three Faroese lakes that deployed XRF data, organic matter (TOC and TN), magnetic susceptibility and $\delta$13C values indicate cooling from 4190 ka BP as judged by higher accumulation rates/increased soil erosion "due to increased influence of e.g., freeze/thaw cycles and thus colder climate" (Olsen et al., 2010).

The possible relationship of these Faroese 4.2 ka BP cooling events to the Hekla 4 eruption (Wastegård, et al., 2018), remains uncertain because the radiocarbon dates (Pilcher et al., 1995) and varve counts (Dörfler et al., 2012) suggest the eruption may have preceded or followed upon the 4.2 ka BP event, but unlikely because "the short residence time of stratospheric sulfate aerosols precludes a lasting influence on the regional energy balance from a single eruption" (Miller et al., 2012:13).

Bradley and Bakke do not mention the three reports of 4.2 ka BP proxy events from the Faroe Islands.

Iceland

The statistical analysis of seven Iceland lake sediment cores documents "episodic glacier expansion between 4.5 and 4.0 ka" (b2k), but "the prominent step toward cooling at 4.5-4.0 ka is statistically indistinguishable from the $\sim$4.2 ka event, and coincides with Hekla 4 (H4), one of the largest explosive eruptions of the Holocene in Iceland" (Giersdóttir et al., 2019). However, "the proxy records from at least these two lakes [SKR and TRK] provide unequivocal evidence for cooling at these times unrelated to tephra-induced soil erosion" (Giersdottir et al 2019). Remarkably, at 4.25 ka BP, the high resolution $\delta$13C spike recorded at Lake Haukadalsvatn, west Iceland (Giersdottir, et al., 2013) is precisely congruent with the high resolution neo-glaciation DBD spike recorded at Lake Hajeren, Svalbard (van der Bilt et al., 2015).

[Figure]

The low resolution regional marine core temperature variability at this time in the northern North Atlantic is noteworthy (Orme et al., 2018: Fig. 7).The Iceland cryosphere expansion is, however, synchronous with cooling events observed at eight high resolution Nordic Seas marine cores:

(1) core MD99-2322 Kangerlussuaq Trough on the east Greenland margin with a CaCO3 spike dated at exceptionally high resolution at 4.2-3.8 ka BP (Stoner et al, 2007: Fig. 11);

(2) core MD99-2269 taken from the Húnaflóaáll Trough on the north Iceland shelf, with a synchronous high resolution CaCO3 spike (Stoner et al, 2007: Fig. 2); both MD99-2322 and MD99-2269 spikes likely from coccolith and formanifera production at surface water cooling (Giraudeau et al, 2004);

3) core MD99-2275 from the shelf of north Iceland providing the 320 diatom sample based SST record, with dating constrained by 15 tephra markers, and recording an abrupt ca. 1 deg C cooling ca. 4200-3800 BP (Jiang et al., 2015);

(4) core MD99-2275, the high resolution chronology marine core off north Iceland, displaying a precipitous alkenone paleothermometry measured 1.6 deg C drop at 4.29 ka BP, followed by a 2.5 deg C drop at 4.16 ka BP that extended for 100 years, and then returned to pre-event levels at 4.0 ka BP. (Jalali et al., 2018);

(5) core MD99-2269 off the North Icelandic Shelf where the biomarker IP25-based sea ice reconstruction "reached its mean value for the entire record at ca 5 cal ka BP, before increasing, continuously, ca 4.3 cal ka BP, broadly in line with the onset of Neoglaciation as seen in some other proxy records (Cabedo-Sanz et al., 2016);

(6) core MD99-2269 off north Iceland recording substantial East Greenland and East Iceland Current changes recorded at ca. 4 ka BP based on diatoms and sediment physical proxies (Moros et al., 2006).

(7) core DS97-2P with an abrupt, 3-stage spike in foraminifera Mg/Ca-derived temperature ca. 4.4 -3.9 ka BP cold event and Sub-Arctic Front alteration at Reykjanes Ridge, south of Iceland at (Moros et al., 2012);

(8) core DA12-11/2-GC01from the south Iceland basin providing the diatom-based SST reconstruction with a pronounced SST cooling from ca. 4 – 2 ka BP, with warmer temperatures prior to 4 ka BP and after 2 ka BP (Orme et al., 2018);

Bradley and Bakke do not mention the abrupt cooling events (1), (2), (3), (4), (5), (6), (7) and conclude ll.119 "None of these [paleoceanographic] records show evidence of an unusual anomaly at 4.2ka B.P.", and ll. 127-128 that their "review of paleoceanographic studies ...provides no evidence for a significant change in major oceanographic conditions that could be linked to the 4.2ka B.P. climate anomaly seen elsewhere."

Greenland lakes, east and west

In eastern Greenland, three lake sediment cores record the abrupt 4.2 ka BP event. At Lake Kulusuk, "at 4.1 ka BP , a sharp increase in XRF- and MS-inferred minerogenic content and decrease in organic matter content indicate the glaciers once again grew large enough to contribute minerogenic material to the lake. The regrowth of the Kulusuk glaciers represents the lowering of the regional snowline" (Balascio et al, 2015). Synchronous hydrologic changes occurred at nearby Flower Valley Lake, where "after 4.1 ka, there is a decrease in evaporative enrichment of the lake water. There is also an abrupt transition to more variable sedimentation marked by sharp increases in magnetic susceptibility, C/N, $\delta$13C, and the concentration of long-chain n-alkanes, showing periodic delivery of terrestrial organic matter and clastic sediment to the lake" (Balascio et al., 2013). Synchronously, the physical and geochemical analyses at Ymer Lake, Ammassalik Island, southeast Greenland, demonstrate a "quiescent Holocene climatic optimum," followed by "Neoglacial cooling, lengthening lake ice cover and shifting wind patterns [that] prompted in-lake avalanching of sediments from 4.2 cal. ka BP onwards" (van der Bilt et al., 2018).

Bradley and Bakke mention Kulusuk, Ymer and Flower Valley lakes, but summarize the

Lake Kulusuk 4.1 ka BP event, ll. 158-160, as "a short-lived 'event' at around that time . . . but this appears to be simply part of the overall deterioration in climate that led to ice growth across the region. There is currently no evidence for a more widespread glacial advance at 4.2ka B.P."

In West Greenland eight lakes have been studied. Jakobshavn region lakes were studied with LOI and MS measurements as well as chironomid-based temperature reconstructions. "Gradual, insolation-driven millennial-scale temperature trends. . . were punctuated by several abrupt climate changes, including a major transient event recorded in all five lakes between 4.3 and 3.2 ka," with a "significant drop in summer temperatures ∼ 4.0 ka BP" (Axford et al., 2013). Earlier, at Braya Sø and Lake E lake organic carbon percentage and LOI spikes at 4.2 ka -3.9 ka BP were identified (D'Andrea et al., 2011). The Lake Lucy record, bolstered with bulk sediment radiocarbon dates, suggests that the western GrIS margin was "near its current margin until ∼4.2 cal ka BP, at which time the ice margin retreated behind Lake Lucy's topographic threshold. The timing of this transition is marked by a steep rise in regional temperatures recorded in the Kangerlussuaq temperature record" (Young and Briner 2015; D'Andrea, et al., 2011)

Bradley and Bakke do not mention the eight west Greenland lakes 4.2 ka BP event proxies.

Greenland and Ellesmere glaciers

In contradistinction to the Swedish, Norwegian, Danish, Faroe Islands, Iceland, and Greenland lacustrine, marine, speleothem, and tree ring data, there are the four glacial core data from Greenland and Ellesmere Island, reviewed from Figure 1:

a-b) Greenland ice sheet total mass balance exhibits a uniquely abrupt 500 Gt/yr reduction at ca 4.5 ka BP and a bounce back at 4.2 ka BP, accompanied by an ice volume reduction in the modeled glacial data (Nielsen et al., 2017);

c) synchronously, NGRIP temperature experienced an abrupt 6.5 deg K degree warm spike at 4.52 – 3.92 ka BP (Gkinis et al., 2014), while SST modeled in the Northwest Atlantic plummeted 4 deg C (Klus et al., 2018). GISP 2 temperature crashed, then rose 2 deg C at ca 4.3 ka BP, while Agassiz and Renland temperatures jumped 2.5 deg C (Vinther et al., 2009);

d) the very high resolution Agassiz, Ellesmere Island 35% melt record (Fisher et al., 2012) congruent with the Lake Hajeren, Svalbard neo-glaciation proxy that spiked five-fold at 4.2 - 4.0 ka BP (van der Bilt, et al., 2015).

Bradley and Bakke, however, claim:

(1) ll. 170-172 "Ice cores from Greenland provide records of past climate variations from oxygen isotopes, glaciochemistry and physical characteristics, which are broadly consistent with those from coastal lake sediments."

(2) ll. 188, the GrIS 4.2 ka BP event was plausibly a "short-lived cooling event, a consequence of the massive eruption of Hekla (in Iceland) at ∼4.2 ka BP."

(3) Figure 3 is GISP2 temperature record, when it is the Agassiz/Renland temperature record (Vinther et al, 2009).

(4) ll. 197 "In summary, there is no compelling evidence for a distinct climatic anomaly at 4.2ka B.P. in ice cores from Greenland."

Linkages

The linkages of these northern North Atlantic 4.2 ka BP events are both extensive and high resolution. The Greenland and Agassiz melt record is synchronous with the 4.2 ka BP event Mt Logan, Yukon ice core melt record, the highest magnitude Holocene event there in the past 4200 years (Fisher et al., 2012), that is in turn linked to especially prominent variations from 4.2 ka BP in the Kuroshio Current, ultimate source of the Yukon westerlies, at the Pulleniatina Event (Zheng et al., 2016), and is precisely synchronous with the Mawmluh Cave record (Berkelhammer et al., 2012). Synchronous,

as well, are adjacent 4.2 ka BP North American aridification event records that stretch from the northwest (Cartier et al., 2018) to the northeast (Newby et al., 2014), to Brazil (Soares Cruz et al., 2019), along Andean South America (e.g., Baker et al, 2009; Schimpf et al., 2011) and to Antarctica (Peck 2015).

The Scandinavian cold and wet records are synchronous with adjacent high resolution Alpine records (e.g., Fohlmeister et al, 2012a, 2012b) and the Urals (Baker et al., 2018), and the adjacent high resolution Mediterranean and West Asian ice cave and speleothem records that extend from Spain (Sancho et al., 2018), Greece (Finne et al 2017), the Levant (Cheng et al., 2015), Iran (Carolin et al., 2019), to the Indian Monsoon domains in the Indian subcontinent (Berkelhammer et al., 2012; Kathayat et al., 2018), and to the East Asian Monsoon domains (e.g., Zhang et al., 2018) and Africa, north to south (e.g., Ruan et al., 2016; Chase et al., 2015) as well. In summary, the northern North Atlantic paleoclimate proxies for the global 4.2 ka BP event comprise high resolution data useful for its eventual global explanation. At this juncture, the authors could 1) test the possible mechanisms by which the northern North Atlantic, with its extensive, coherent, and high resolution records, was disconnected from the global climate system at 4.2 ka BP, or 2) test the possible mechanisms by which it was connected.

Conclusion

A recent synthesis for the Arctic concluded that "acceleration of cooling ca. 4.2 ka is uncommon, with a notable (but nonsignificant) peak in cooling onset probability around that time found only in Greenland" (McKay et al 2018). That conclusion, however, was derived from a 2014 compilation (Sundquist et al., 2014) with few updates, and is both out-of-date and erroneous. The Bradley and Bakke "Northern North Atlantic" article that is proposed for CP, concludes ll. 243-244, 248-251, that "A review of paleoceanographic and terrestrial paleoclimatic data from around the northern North Atlantic reveals no compelling evidence for a significant climatic anomaly at ∼4.2ka B.P....Although a few records do show a distinct anomaly around 4.2ka B.P. (associated with a glacial advance), this is not widespread and we interpret it as a local signal of the overall climatic deterioration that characterized the late Holocene."

Bradley and Bakke ignore, however, the 4.2 ka BP event data from Svalbard, Sweden, Norway, Denmark, Faroes Islands, Iceland, west Greenland, and the relevant Nordic Seas marine core data, and misrepresent the elevation-corrected Greenland Ice Sheet data, the Agassiz ice core data, and the coincidence of northern North Atlantic 4.2 ka BP event glacial melt and lake cooling. In summary, the proposed article (a) ignores most of the data reviewed here for the 4.2 ka BP event in the northern North Atlantic, (b) misrepresents data in the few cases that are discussed, and (c) fails to identify the regionally coherent feature of the 4.2 ka BP event in the northern North Atlantic: abrupt lacustrine, marine and terrestrial cooling synchronous with elevation-corrected abrupt glacial warm events, as represented in Figure 1. The Bradley and Bakke proposed article does not approach the consensual standards for science publication.

References

Andresen, C., Björck, S., Rundgren, M., Conley, D., and Jessen, C.: Rapid Holocene climate changes in the North Atlantic: evidence from lake sediments from the Faroe Islands, Boreas, 35, 23-34, 2006.

Antonsson, K. and Seppä, H.: Holocene temperatures in Bohuslän, southwest Sweden: a quantitative reconstruction from fossil pollen data, Boreas, 36, 400-410, 2007.

Axford, Y., Losee, S., Briner, J. P., Francis, D. R., Langdon, P. G., and Walker, I. R.: Holocene temperature history at the western Greenland Ice Sheet margin reconstructed from lake sediments, Quaternary Sci Rev, 59, 87-100, 2013.

Bailey, H. L., Kaufman, D. S., Sloane, H. J., Hubbard, A. L., Henderson, A. C. G., Leng, M. J., Meyer, H., and Welker, J. M.: Holocene atmospheric circulation in the central North Pacific: A new terrestrial diatom and $\delta$18O dataset from the Aleutian Islands, Quaternary Sci Rev, 194, 27-38, 2018.

Baker, Jonathan L., M, Lachniet, S., Chervyatsova, O., Asmerom, Y., Polyak, V.J.: Holocene warming in western continental Eurasia driven by glacial retreat and greenhouse forcing, Nature Geoscience 10, 430–435, 2017.

Bakke, J., Dahl, S. O., Paasche, Ø., Riis Simonsen, J., Kvisvik, B., Bakke, K., and Nesje, A.: A complete record of Holocene glacier variability at Austre Okstindbreen, northern Norway: an integrated approach, Quaternary Sci Rev, 29, 1246-1262, 2010.

Balascio, N.J., D'Andrea, W.J. and Bradley, R.S.: Glacier response to North Atlantic climate variability during the Holocene, Climate of the Past, 11, 1587-1598, 2015.

Balascio, N.L., D'Andrea, W.J., Bradley, R.S., Perren, B.: Biogeochemical evidence for hydrologic changes during the Holocene in a lake sediment record from southeast Greenland, The Holocene 23, 1428–1439, 2013.

Berkelhammer, M. Sinha, A., Stott, L., Cheng, H., Pausata, F.S.R., and Yoshimura, K.: An Abrupt Shift in the Indian Monsoon 4000 Years Ago, in Climates, Landscapes, and Civilizations, Geophysical Monograph Series 198. 10.1029/2012GM001207, 2012.

Blair, C. L., Geirsdóttir, Á., and Miller, G. H.: A high-resolution multi-proxy lake record of Holocene environmental change in southern Iceland, J Quaternary Sci, 30, 281-292, 2015.

Briner, J. P., McKay, N. P., Axford, Y., Bennike, O., Bradley, R. S., de Vernal, A., Fisher, D., Francus, P., Fréchette, B., Gajewski, K., Jennings, A., Kaufman, D. S., Miller, G., Rouston, C., and Wagner, B.: Holocene climate change in Arctic Canada and Greenland, Quaternary Sci Rev, 147, 340-364, 2016.

Cabedo-Sanz, P., Belt, S.T., Jennings, A.E., Andrews, J.T., and Geirsdóttir, Á.: Variability in drift ice export from the Arctic Ocean to the North Iceland Shelf over the last 8000 years: A multi-proxy evaluation, Quat. Sci. Rev., 146, 99-115, 2016.

Carolin, Stacy A., Walker, Richard T., Day, Christopher C., Ersek Vasile, Sloan, R. Alastair, Dee, Michael W., Talebian, Mortezan, and Henderson, Gideon M.: Precise

timing of abrupt increase in dust activity in the Middle East coincident with 4.2 ka social change, Proc Natl Acad Sci, 116, 67-72, 2019.

Carter, V. A., Shinker, J. J., and Preece, J.: Drought and vegetation change in the central Rocky Mountains and western Great Plains: potential climatic mechanisms associated with megadrought conditions at 4200 cal yr BP, Clim. Past, 14, 1195-1212, https://doi.org/10.5194/cp-14-1195-2018, 2018.

Cheng, H., Sinha, A., Verheyden, S., Nader, F. H., Li, X. L., Zhang, P. Z., Yin, J. J., Yi, L., Peng, Y. B., Rao, Z. G., Ning, Y. F., and Edwards, R. L.: The climate variability in northern Levant over the past 20,000 years, Geophys Res Lett, 42, 8641-8650, 2015.

D'Andrea, William J., Huang, Yongsong, Fritz, Sherilyn C. and Anderson, N. John: Abrupt Holocene climate change as an important factor for human migration in West Greenland, Proc Natl Acad Sci 108, 9765–9769, 2011.

Carter, V.A. and Shinjker, Jacqueline: Drought and vegetation change in the central Rocky Mountains: Potential climatic mechanisms associated with the mega drought at 4200 cal yr BP. CoP Clim. Past Discuss., https://doi.org/10.5194/cp-2017-107, 2017.

Chase, B. M., Lim, S., Chevalier, M., Boom, A., Carr, A. S., Meadows, M. E., and Reimer, P. J.: Influence of tropical easterlies in southern Africa's winter rainfall zone during the Holocene, Quaternary Sci Rev, 107, 138-148, 2015.

D'Andrea, William J., Huang, Yongsong, Fritz, Sherilyn C. and Anderson, N. John: Abrupt Holocene climate change as an important factor for human migration in West Greenland, Proc Natl Acad Sci 108, 9765–9769, 2011.

Dörfler, Walter, Feeser, Ingo, van den Bogaard, Christel, Dreibrodt, Stefan, Erlenkeuser, Helmut, Kleinmann, Angelika, Merkt, Josef, Wiethold, Julien: A high-quality annually laminated sequence from Lake Belau, Northern Germany: Revised chronology and its implications for palynological and tephrochronological studies, The

Holocene 22, 1413–1426, 2012.

Eckstein, J., Leuschner, H. H., Giesecke, T., Shumilovskikh, L., and Bauerochse, A.: Dendroecological investigations at Venner Moor (northwest Germany) document climate-driven woodland dynamics and mire development in the period 2450–2050 BC, The Holocene, 20, 231-244, 2010.

Edvardsson, J.: Mid- to Late Holocene climate transition and moisture dynamics inferred from South Swedish tree-ring data, Journal of Quaternary Sci, 31, 256-264, 2016.

Finné, M, Holmgren K, Shen C-C, Hu H-M, Boyd M, Stocker S: Late Bronze Age climate change and the destruction of the Mycenaean Palace of Nestor at Pylos. PLoS ONE 12,12:e0189447. https://doi.org/10.1371/journal.pone.0189447, 2017.

Fisher, David, Zheng, J., Burgess, D., Zdanowicz, C., Kinnard, C., Sharp, M., Bourgeois, J.: Recent melt rates of Canadian arctic ice caps are the highest in four millennia, Global and Planetary Change, 84–85, 3-7, 2012.

Fohlmeister, J., Schröder-Ritzrau, A., Scholz, D., Spötl, C., Riechelmann, D. F. C., Mudelsee, M., Wackerbarth, A., Gerdes, A., Riechelmann, S., Immenhauser, A., Richter, D. K., and Mangini, A.: Bunker Cave stalagmites: an archive for central European Holocene climate variability, Clim Past, 8, 1751-1764, 2012a.

Fohlmeister, J., Vollweiler, N., Spötl, C., and Mangini, A.: COMNISPA II: Update of a mid-European isotope climate record, 11 ka to present, The Holocene, 23, 749-754, 2012b.

Geirsdóttir, Á., Miller, G. H., Larsen, D. J., and Ólafsdóttir, S.: Abrupt Holocene climate transitions in the northern North Atlantic region recorded by synchronized lacustrine records in Iceland, Quaternary Sci Rev, 70, 48-62, 2013.

Geirsdóttir, Áslaug, Miller, Gifford H., Andrews, John T., Harning, David J., Anderson, Keif F., Florian, Christopher, Larsen, Darren J., Thordarson, Thor: The onset of

neoglaciation in Iceland and the 4.2 ka event, Clim. Past, 15, 25-40, 2019.

Giraudeau, J., Jennings, A.E., Andrews, J.T.: Timing and mechanisms of surf ace and intermediate water circulation changes in the Nordic Seas over the last 10,000 cal years: a view from the North Iceland shelf, Quaternary Science Reviews 23, 2127–2139, 2004.

Gkinis, V., Simonsen, S.B., Buchardt, S.L., White, J.W.C., Vinther, B.M.: Water isotope diffusion rates from the NorthGRIP ice core for the last 16,000 years – Glaciological and paleoclimatic implications, Earth and Planetary Science Letters 405, 132-141, 2014.

Goslin, J., Fruergaard, M., Sander, L., Galka, M., Menviel, L., Monkenbusch, J., Thibault, N., and Clemmensen, L. B.: Holocene centennial to millennial shifts in North-Atlantic storminess and ocean dynamics, Sci Rep, 8, 12778, 2018.

Gunnarson, B.E.: Temporal distribution pattern of subfossil pines in central Sweden: perspective on Holocene humidity fluctuations, The Holocene, 18, 69-77, 2008.

Hammarlund, D., Björck, S., Buchardt, B., Israelson, C., and Thomsen, C. T.: Rapid hydrological changes during the Holocene revealed by stable isotope records of lacustrine carbonates from Lak Igelsjön, southern Sweden, Quarternary Sci Rev, 22, 353-370, 2003.

Jalali, Bassem, Sicre, Marie-Alexandrine, Azuara, Julien, Pellichero, Violaione, Combourieu-Nebout, Nathalie: Influence of the North Atlantic subpolar gyre circulation on the 4.2 ka BP event, Clim. Past, https://doi.org/10.5194/cp-2018-159, 2018.

Jiang, H., Muscheler, R., Björck, S., Seidenkrantz, M.-S., Olsen, J., Sha, L., Sjolte, J., Eiríksson, J., Ran, L., Knudsen, K.-L., and Knudsen, M.F.: Solar forcing of Holocene summer sea-surface temperatures in the northern North Atlantic, Geology, 43,2-5, 2015.

Kathayat, Gayatri, Cheng, Hai, Sinha, Ashish, Berkelhammer, Max, Zhang, Haiwei, Duan, Pengzhen, Li, Hanying, Li, Xianglei, Ning, Youfeng, and Edwards, R. Lawrence

Edwards: Evaluating the timing and structure of the 4.2 ka event in the Indian summer monsoon domain from an annually resolved speleothem record from Northeast India, Clim. Past, 14, 1869-1879, 2018.

Kobashi, Tazkuro, Menviel, L., Jeltsch-Thömmes, A., Vinther, B.M., Box, J.E., Muscheler, R., Nakaegawa, T., Pfister, P.L., Döring, M., Leuenberger, M., Wanner, H., Ohmura, A.: Volcanic influence on centennial to millennial Holocene Greenland temperature change, Scientific Reports, 7: 1441, 2017.

Klus, A., Prange, M., Varma, V., Tremblay, L. B., and Schulz, M.: Abrupt cold events in the North Atlantic Ocean in a transient Holocene simulation, Clim. Past, 14, 1165-1178, 2018.

Larsen, D. J., Miller, G. H., Geirsdóttir, Á., and Ólafsdóttir, S.: Non-linear Holocene climate evolution in the North Atlantic: a high-resolution, multi-proxy record of glacier activity and environmental change from Hvítárvatn, central Iceland, Quaternary Sci Rev, 39, 14-25, 2012.

Lauritzen, Stein-Erik and Joyce Lundberg: Calibration of the speleothem delta function: an absolute temperature record for the Holocene in northern Norway. The Holocene 9, 659–669, 1999.

Lecavalier, Benoit S., Fisher, David A., Milne, Glenn A., Vinther, Bo M., Tarasov, Lev, Huybrechts, Philippe, Lacelle, Denise, Main, Brittany, Zheng, James, Bourgeois, Joce-lyne, Dykeh, Arhtur S.: High Arctic Holocene temperature record from the Agassiz ice cap and Greenland ice sheet evolution, Proc Natl Acad Sci, 114, 5952-5957, 2017.

McKay, Nicholas P., Kaufman, D.S., Routson, C.C., Erb, M.P., Zander, P.D.: The Onset and Rate of Holocene Neoglacial Cooling in the Arctic, Geophysical Research Letters, 45, 12487–12496, 2018.

Miller, Gifford H, Geirsdóttir, A., Zhong, Y., Larsen, D.J., Otto-Bliesner, B.L., Holland, M.M., Bailey, D.A., Refsnider, K.A., Lehman, S.J., Southon, J.R., Anderson, C., Björnsson, H., Thordarson, T.: Abrupt onset of the Little Ice Age triggered by volcanism and sustained by sea-ice/ocean feedbacks, Geophysical Research Letters, 39, 2012.

Moros, Matthias, Andrews, J.T., Eberl, D.D., Jansen, E.: Holocene history of drift ice in the northern North Atlantic:Evidence for different spatial and temporal modes, Paleoceanography 21, PA2017, doi:10.1029/2005PA001214, 2006.

Moros, Matthias, Jansen, E., Oppo, D.W., Giraudeau, J., Kuijpers, A.: Reconstruction of the late-Holocene changes in the Sub-Arctic Front position at the Reykjanes Ridge, north Atlantic, The Holocene 22, 877–886, 2012.

Newby, Paige E., N. Shuman, Bryan, Donnelly, Jeffery P., Karnauskas, Kristopher B. and Marsicek, Jeremiah: Centennial-to-millennial hydrologic trends and variability along the North Atlantic Coast, USA, during the Holocene. GRL 10.1002/2014GL060183, 2014.

Nielsen, Lisbeth T., AÃřalgeirsdóttir, GuÃřfinna, Gkinis, Vasileos, Nuterman, R., Hvidberg, C.S.: The effect of a Holocene climatic optimum on the evolution of the Greenland ice sheet during the last 10 kyr, Journal of Glaciology 64, 477–488, 2018.

Olsen, Jesper, S. Björck, M. J. Leng, E.R. Gudmundsdóttir, B.V. Odgaard, C. M. Lutz, C. P. Kendrick, T. J. Andersen, M.-S. Seidenkrantz: Lacustrine evidence of Holocene environmental change from three Faroese lakes: a multiproxy XRF and stable isotope study, Quaternary Sci Rev, 29, 276-2780, 2010.

Orme, L. C., A. Miettinen, D. Divine, K. Husum, C. Pearce, N. Van Nieuwenhove, A. Born, R. Mohan, M.-S. Seidenkrantz, Subpolar North Atlantic sea surface temperature since 6 ka BP: Indications of anomalous ocean-atmosphere interactions at 4-2 ka BP, Quaternary Sci Rev, 194, 128-142, 2018.

Peck, V. L., Allen, C. S., Kender, S., McClymont, E. L., and Hodgson, D. A.: Oceanographic variability on the West Antarctic Peninsula during the Holocene and the influence of upper circumpolar deep water, Quaternary Sci Rev, 119, 54-65, 2015.

Perner, K., M. Moros, E. Jansen, A. Kuijpers, S.R. Troelstra, M.A. Prins: Subarctic Front migration at the Reykjanes Ridge during the mid- to late Holocene: evidence from planktic foraminifera, Boreas, 47, 175-188, 2018.

Pilcher, J.R., Hall, V.A., McCormac F.G.: Dates of Holocene Icelandic volcanic eruptions from tephra layers in Irish peats, The Holocene 5, 103-110, 1995.

Rassmussen, Tine L. and Thomsen, Erik: Holocene temperature and salinity variability of the Atlantic Water inflow to the Nordic seas, The Holocene 20, 1223–12, 2010.

Repschläger, J., D. Garbe-Schönberg, M. Weinelt, R. Schneider: Holocene evolution of the North Atlantic subsurface transport, Clim Past, 13, 333-344, 2017.

Risebrobakken, B., Dokken, T., Smedsrud, L. H., Andersson, C., Jansen, E., Moros, M., and Ivanova, E. V.: Early Holocene temperature variability in the Nordic Seas: The role of oceanic heat advection versus changes in orbital forcing, Paleoceanography, 26, 2011.

Ruan, J., Kherbouche, F., Genty, D., Blamart, D., Cheng, H., Dewilde, F., Hachi, S., Edwards, R. L., Regnier, E., and Michelot, J. L.: Evidence of a prolonged drought ca. 4200 yr BP correlated with prehistoric settlement abandonment from the Gueldaman GLD1 Cave, Northern Algeria, Clim Past, 12, 1-14, 2016.

Sancho, C., Belmonte, Á., Bartolomé, M., Moreno, A., Leunda, M., and López-Martínez, J.: Middle-to-late Holocene palaeoenvironmental reconstruction from the A294 ice-cave record (Central Pyrenees, northern Spain), Earth Planet Sc Lett, 484, 135-144, 2018.

Schimpf, D., Kilian, R., Kronz, A., Simon, K., Spötl, C., Wörner, G., Deininger, M., and Mangini, A.: The significance of chemical, isotopic, and detrital components in three coeval stalagmites from the superhumid southernmost Andes (53°S) as high-resolution palaeo-climate proxies, Quaternary Sci Rev, 30, 443-459, 2011.

Soares Cruz, A. P., Fernandes Barbosa, C., Blanco, A. M., de Oliveira, C. A., Guizan
Silva, C., and Sícoli Seoane, J. C.: Mid-Late Holocene event registered in organo-siliciclastic-sediments of Lagoa Salgada carbonate system, Southeast Brazil, Clim. Past Discuss., https://doi.org/10.5194/cp-2019-27, in review, 2019.

Stoner J.S., Jennings A.E., Kristjánsdóttir G.B., Dunhill, G., Andrews, J.T., and Hardardóttir, J.: A paleomagnetic approach toward refining Holocene radiocarbon based chronostratigraphies: Paleoceanographic records from North Iceland (MD99-2269) and East Greenland (MD99-2322) margins, Paleoceanography, 22, PA1209, 2007.

Sundqvist, H. S., Kaufman, D. S., McKay, N. P., Balascio, N. L., Briner, J. P., Cwynar, L. C., Sejrup, H. P., Seppä, H., Subetto, D. A., Andrews, J. T., Axford, Y., Bakke, J., Birks, H. J. B., Brooks, S. J., de Vernal, A., Jennings, A. E., Ljungqvist, F. C., Rühland, K. M., Saenger, C., Smol, J. P., and Viau, A. E.: Arctic Holocene proxy climate database – new approaches to assessing geochronological accuracy and encoding climate variables, Clim. Past, 10, 1605-1631, https://doi.org/10.5194/cp-10-1605-2014, 2014.

van der Bilt, W. G. M., Bakke, J., Vasskog, K., D'Andrea, W. J., Bradley, R. S., and Ólafsdóttir, S.: Reconstruction of glacier variability from the lake sediments reveals dynamic Holocene climate in Svalbard, Quaternary Sci Rev, 126, 201-218, 2015.

van der Bilt, W. G. M., D'Andrea, W. J., Bakke, J., Balascio, N. L., Werner, J. P., Gjerde, M., and Bradley, R. S.: Alkenone-based reconstructions reveal four-phase Holocene temperature evolution for High Arctic Svalbard, Quaternary Sci Rev, 183, 204-213, 2018a.

van der Bilt, W. G. M., Rea, B., Spagnolo, M., Roerdink, D. L., Jørgensen, S. L., and Bakke, J.: Novel sedimentological fingerprints link shifting depositional processes to Holocene climate transitions in East Greenland, Global Planet Change, 164, 52-64, 2018b.

Vinther, B., Buchardt, S.L., Clausen, H.B., Dahl-Jensen, D., Johnsen, S.J., Fisher, D.A.,

Koerner, R.M., Raynaud, D., Lipenkov, V., Andersen, K.K., Blunier, T., Rasmussen, S.O., Steffensen, J.P. and Svensson, A.M.: Significant Holocene thinning of the Greenland ice sheet, Nature, 515, 385-388, 2009.

Wastegård, S.,Gudmundsdóttir, E.R., Lind, E.M., Timms, R.G.O., Björck, S., Hannon, G.E., Olsen, J., Rundgren, M.: Towards a Holocene tephrochronology for the Faroe Islands, North Atlantic, Quaternary Science Reviews 195, 195-214, 2018.

Young, N. E. and Briner, J. P.: Holocene evolution of the western Greenland Ice Sheet: Assessing geophysical ice-sheet models with geological reconstructions of ice-margin change, Quaternary Sci Rev, 114, 1-17, 2015.

Zhang, N., Yang, Y., Cheng, H., Zhao, J., Yang, X., Liang, S., Nie, X., Zhang, Y., and Edwards, R. L.: Timing and duration of the East Asian summer monsoon maximum during the Holocene based on stalagmite data from North China, The Holocene, 28, 1631-1641, 2018.

Zheng, Xufeng, Li, S. J., Kao, X. Gong, M., Frank, G., Kuhn, W. Cai, H., Yang, S., Wan, H., Zhang, F., Jiang, E., Hathorne, Chen, Z.,. Hui, B.: Synchronicity of Kuroshio Current and climate system variability since the Last Glacial Maximum, Earth and Planetary Science Letters 452, 247-257, 2016.

Please also note the supplement to this comment:
https://www.clim-past-discuss.net/cp-2018-162/cp-2018-162-RC2-supplement.pdf

[Figure]

**Fig. 1.** Fig 1 northern North Atlantic Weiss 2019 CP

---

## Author Comment (AC1) · 1 Apr 2019

We find the re-interpretation of papers that Weiss has attempted (many of which we wrote ourselves, or on which we were co-authors) quite a remarkable effort in revisionism. But a passionate defense of a paradigm in which one is heavily invested is not a sufficient reason to dismiss our review of the literature. We address the individual points raised by Weiss below, but first we offer some general comments.

The title of the Special Issue to which our paper was submitted, is "The 4.2ka B.P. EVENT". So, what is "an event"? The word derives from the Latin, "eventus" meaning a singular happening, an occurrence, an incident. It clearly implies something that has a beginning and an end, otherwise it would not be an event. And, in the case of the Special Issue, this event must be dated at 4.2ka B.P. It is abundantly clear that there was a shift in climate between 5000 and 4000 years BP, leading to a period of neoglaciation (cf. McKay et al., 2018). The subsequent millennia witnessed a series of climatic oscillations that led to the advance and retreat of alpine glaciers, culminating in the most widespread advances, which occurred during the last millennium. As neoglaciation set in, it is no surprise that some glaciers advanced around 4.2ka B.P. (as in the case of Kulusk, SE Greenland and at Hajaren, Svalbard where we have worked) while others advanced somewhat earlier, and others somewhat later. But one cannot ignore the fact that there was an underlying, systematic change in climate beginning around ∼5ka B.P. Around the North Atlantic, this change in climate involved a cooling trend, and multi-decadal to centennial fluctuations were superimposed on that trend. Cherry-picking records that happen to show a multi-decadal anomaly around 4.2ka B.P. (or at other times before or after that time) in order to promote the notion of a global-scale climate disturbance is not good scientific practice. Yet too often those who seek to memorialize the 4.2ka B.P. time period have brushed aside chronological uncertainties, low resolution records and inconsistent proxies in order to promote a story that does not always stand up to scrutiny. Such misguided efforts cast further doubt on the highly dubious argument for a late Holocene boundary at 4.2ka B.P., which Weiss has helped to promote (Walker et al., 2018). We have little doubt that some proxy records provide strong evidence for a short-lived climatic anomaly (often a drought) centered around 4.2ka B.P. But many other locations do not show such evidence. If we are going to understand what may have caused "the 4.2ka B.P. event" in some areas, we first need to define which geographical areas were affected and do so objectively and without bias using well resolved records with good dating control. A good first step toward such a solution has been made by Sundqvist et al. (2014).

Specific comments: As our paper clearly noted, "we review sedimentary records from the northern North Atlantic (north of 60°N) with a focus on evidence for an "event" around 4.2ka B.P. ". Obviously, that does not include research in Denmark, southern Sweden, Ellesmere Island or West Greenland so we will not address comments on

papers related to those areas. Greenland: The paper by Nielsen et al (2018) uses a model with different assumptions about the temperature and accumulation history of the Greenland Ice Sheet to derive mass balance estimates over time. Only 2 of the 6 simulations generate a signal at ∼4.2ka B.P. Both of those experiments are based on data from Gkinis et al. (2014) so one should not be surprised to see a correlation between those results. The Gkinis et al. diffusion model does generate a large (4-5C) positive temperature anomaly around 4.2ka B.P. in NGRIP, which is not seen in the Vinther et al. (2009) paleotemperature reconstructions, or indeed in any other Greenland ice core isotopic records (Fig. R1). In those records, variations in isotopes (and derived temperatures) are all relatively small (e.g.  <1C in the Vinther et al., record, and that change mainly results from the Agassiz Ice core, central Ellesmere Island). Gkinis et al. derive their result by applying a coupled water isotope diffusion and firn densification model, with the GICC05 chronology and an estimate of the total thinning function obtained from another flow model. Given the assumptions made in this model, it would be prudent to see if the result can be reproduced in other records; such a large anomaly would almost certainly be recorded in temperature sensitive proxies, such as chironomid data in coastal lakes. So far, no such confirmatory data exist (cf. Axford et al., 2017). Kobashi et al ( 2017) also provide no support for such a large anomaly at 4.2ka B.P. as that derived by Gkinis et al (2014). Furthermore, the advance of ice in southeast Greenland at the same time (Kulusuk Lake; Balascio et al., 2015), seems quite at odds with such an exceptionally large positive temperature anomaly on the Greenland Ice Sheet. So, while we are intrigued by the results of Gkinis et al.,(2014) we are not yet convinced. Nevertheless, for the sake of completeness, we were remiss in not mentioning their reconstruction in our summary. Svalbard. The van der Bilt (2015) evidence for an ice advance was mentioned in our paper. van der Bilt et al (2018a) indicate a drop in temperature of ∼2C at 4.2ka B.P. in Lake Hajeren. However, there is no comparable event signal in the adjacent valley where glaciers advanced later in the Holocene (Røthe et al., 2017). Gjerde et al. (2018) was not "misrepresented". Luoto et al (2017) reveal no warming at 4.2ka B.P., nor does Balascio

et al. (2018). I note here that both Bradley and Bakke were co-authors on all of these papers (except Luoto et al) and conducted fieldwork in all of the sites mentioned so it is somewhat bizarre to find our results being reinterpreted by Weiss. Such efforts say more about the underlying motivation, which is to find something everywhere, when it may not exist. We have tried to be more objective.

In the Faroe Islands (where Bradley and Bakke have also worked) Andressen et al find a drop in silica and increase in clastic material at 4.2ka B.P., which are comparable in magnitude to earlier periods. They assign a zone boundary to 4.2ka B.P. but do not recognize this as an event; rather, it signals a change in sedimentary regime (S4) which continued to ∼2000 B.P. Similarly, the Olsen et al. (2010) and Rasmussen et al. (2010) papers recognize a change at ∼4ka B.P. which indicates the onset of neoglacial conditions in the archipelago (Olsen et al.'s Zone 8). This conforms to our overall conclusions.

The Icelandic evidence has been reviewed by Geirsdottir et al. (2019) and as we noted, we did not attempt to duplicate that paper. Now that it has been accepted, we will reference its conclusions in revisions to our paper.

Additional papers cited:

Axford, Y., Levy, L.B., Kelly, M.A., Francis, D.R., Hall, B.L., Langdon, P.G. and Lowell, T.V., 2017. Timing and magnitude of early to middle Holocene warming in East Greenland inferred from chironomids. Boreas, 46(4), pp.678-687. McKay, N.P., Kaufman, D.S., Routson, C.C., Erb, M.P. and Zander, P.D., 2018. The onset and rate of Holocene Neoglacial cooling in the Arctic. Geophysical Research Letters, 45(22), pp.12-487. Sundqvist, H. S., Kaufman, D. S., McKay, N. P., Balascio, N. L., Briner, J. P., Cwynar, L. C., et al. (2014). Arctic Holocene proxy climate database—New approaches to assessing geochronological accuracy and encoding climate variables. Climate of the Past, 10, 1605–1631. Walker, Mike JC, Max Berkelhammer, Svante Björck, Les C. Cwynar, David A. Fisher, Antony J. Long, John J. Lowe, Rewi M. Newnham, Sune O.

Rasmussen, and Harvey Weiss. "Formal subdivision of the Holocene Series/Epoch: a Discussion Paper by a Working Group of INTIMATE (Integration of ice‐core, marine and terrestrial records) and the Subcommission on Quaternary Stratigraphy (International Commission on Stratigraphy)." Journal of Quaternary Science, 27, no. 7 (2012): 649-659.

Please also note the supplement to this comment:
https://www.clim-past-discuss.net/cp-2018-162/cp-2018-162-AC1-supplement.pdf
* * *

---

## Short Comment (SC2) · 2 Apr 2019

**Comment on H. Weiss' review of "Is there evidence for a 4.2 ka B.P. event in the northern North Atlantic region?"**

Z. Bora ÖN [*1], Alan M. GREAVES[2], Sena AKÇER-ÖN[1], and M. Sinan ÖZEREN[3]

[1] *Muğla SK Üniversitesi, Jeoloji Mühendisliği Bölümü, 48000, Muğla, Türkiye*
[2] *University of Liverpool, Department of Archaeology, Liverpool, United Kingdom*
[3] *İTÜ, Avrasya Yer Bilimleri Enstitüsü, 34469, Ayazağa, İstanbul, Türkiye*

April 2, 2019

In their manuscript, Bradley and Bakke (2019) compile marine and terrestrial records from the North Atlantic, north of $60°$ N, to discuss the existence or otherwise of an abrupt climatic change around 4.2 ka BP across this extensive region. Their discussion is mainly based on data which do not conform to that expected of such an abrupt event. According to Weiss, as the *confirming evidences* of an abrupt climatic change are not cited in the manuscript, their article therefore "does not approach the consensual standards for science publication".

While confirming evidences might at first appear to strengthen a hypothesis, we believe that deeper discussions that test the inherent robustness of any theory should embrace counter-evidence and address antithetical points of view. A "good" scientific model should explain all existing observations, and it must rule out certain things, i.e. it must be prohibitive (Popper, 1963; Hawking and Mlodinow, 2010). In the geological sciences, dating inaccuracies, measurement errors, dependency of proxy data on different processes, and/or inappropriate interpretations of proxy data can all be problematic
* * *
[*] boraon@mu.edu.tr

when seeking conclusive evidence. The problematic nature of such data means that one can lightly dismiss any non-confirming studies when they arise from such false auxiliary assumptions. However, the same reasoning can also be applied to the supporting evidence itself (cf. Berkelhammer et al., 2013; Kathayat et al., 2018).

As Popper (1963) stated, obtaining confirmations of a theory is easy, if one looks for it. Therefore, the idea that to strengthen the scientific status of a theory one should list further confirmatory evidence is, for Popper, merely an illusion. It is our view that discussions on the spatial coverage of the "4.2 ka BP event" must address the contradictory examples and even test "confirming" evidences if its robustness as a hypothesis is to be accepted. By doing so we may then modify the theory in a nuanced manner that incorporates the physical basis behind it or, potentially, even refuse it altogether. Again, as Popper (1963, p. 48) states: "A theory which is not refutable by any conceivable event is non-scientific. Irrefutability is not a virtue of a theory (as people often think) but a vice."

**References**

Berkelhammer, M., Sinha, A., Stott, L., Cheng, H., Pausata, F.S.R., K., Y., 2013. An Abrupt Shift in the Indian Monsoon 4000 Years Ago, in: Giosan, L., Fuller, D.Q., Nicoll, K., Flad, R.K., Clift, P.D. (Eds.), Climates, Landscapes, and Civilizations. American Geophysical Union, pp. 75–88. doi:10.1029/2012GM001207.

Bradley, R., Bakke, J., 2019. Is there evidence for a 4.2 ka BP event in the northern North Atlantic region? Climate of the Past Discussions 2019, 1–22. doi:10.5194/cp-2018-162.

Hawking, S., Mlodinow, L., 2010. The Grand Design. Random House Publishing Group.

Kathayat, G., Cheng, H., Sinha, A., Berkelhammer, M., Zhang, H., Duan, P., Li, H., Li, X., Ning, Y., Edwards, R.L., 2018. Evaluating the timing and structure of the 4.2 ka BP event in the Indian summer monsoon domain from an annually resolved speleothem record from Northeast India. Climate of the Past 14, 1869–1879. doi:10.5194/cp-14-1869-2018.

Popper, K.R., 1963. Conjectures and Refutations: The Growth of Scientific Knowledge. Classics Series. 2002 reprint from the 3$^{rd}$ ed., Routledge.

---

## Referee Comment (RC3) · Harvey Weiss (Referee) · 4 Apr 2019

Karl Popper, Imre Lakatos, and the 4.2 ka BP event in the northern North Atlantic, Anatolia and the Indus

Harvey Weiss1

1 School of Forestry and Environmental Studies, Yale University

What do we learn from Bradley and Bakke (2019)?

Bradley and Bakke (2019) state, "A review of paleoceanographic and terrestrial paleo-climatic data from around the northern North Atlantic reveals no compelling evidence for a significant climatic anomaly at ∼4.2ka B.P.", based upon their evaluation of their

sample of northern North Atlantic proxies publications. They did not, however, consider about 50 available and relevant high resolution proxies, most of which document abrupt 4.2 ka BP events in Svalbard, Norway,Sweden, Denmark, Faroe Islands, Iceland, Greenland, and Ellesmere Island, and their proposed article, therefore, "does not approach the consensual standards for science publication" (Weiss 2019). Ön et al. (2019) proclaim jejunely that good science includes all data, that which supports and that which falsifies a theory, and remind us that Karl Popper (1962) claimed falsifiability as the test of scientific verisimilitude and veracity – though Kuhn (1962: 77) had advised, "No process yet disclosed by the historical study of scientific development at all resembles the methodological stereotype of falsification by direct comparison with nature." Similarly, Imre Lakatos (1978) famously revisited Popper's claim with his "Methodology of Scientific Research Programmes," pointing out that Popper's criterion is too restrictive and would invalidate much of everyday scientific practice. Popper's falsification demarcation has since been challenged and dismissed repeatedly (e.g., Hansson, 2006).

Skewed and truncated sampling meets the science standards of neither Popper nor Lakatos. Bradley and Bakke (2019) show it is easy to falsify a scientific claim by ignoring all the positive evidence. Surely, a meta-analytic study is one desideratum for further 4.2 ka BP event study but, of course, the selection criteria remain labile.

What do we learn from Lake Hazar, Anatolia?

The northern North Atlantic region is a small, but intensively studied, part of the Eurasian climate systems that are driven by the westerlies, the Indian Monsoon and the East Asian Monsoon. Ön et al (2018) believe that their Hazar Lake, Turkey sediment core disproves the 4.2-3.9 ka BP megadrought observations recorded globally, including in west Asia and its Anatolian plateau (Turkey). That is, Ön et al., (2018) believe they have provided the Anatolian Popperian falsifiability test for 4.2-3.9 ka BP megadrought across west Asia. Fig. 1 lists the twelve Anatolian paleoclimate proxies for the 4.2 ka BP event. Two of the twelve proxy analyses report wet conditions at 4.2

ka BP. Sofular Cave (Göktürk et al., 2011) is, however, not a proxy for the Mediter-ranean westerlies; it represents the unique situation of the Black Sea and its intrusive, micro-region, northern precipitation. The only westerlies proxy analysis publication for Anatolia that presents a wet 4.2 ka BP period is the Ön et al (2018) Hazar Lake study that statistically analyses XRF data. The Hazar Lake sediment core, however, (a) has a sampling resolution of 175 years, (b) is constrained by two calibrated radiocarbon dates separated by four to five thousand years, and (c) is misrepresented by Ön et al., (2018). The $\delta13C$ and $\delta18O$ values at Lake Hazar record the 4.2 ka BP abrupt megadrought event (Fig. 2) – despite their study's poor resolution. In central Anatolia, the 4.2 ka BP event is recorded at Nar Gölü with 5 year sampling resolution (Dean et al., 2015), correcting the earlier Roberts et al. (2001) publication. The Anatolian 4.2 ka BP event data listed in Fig. 1 are sandwiched between the westerlies' 4.2 ka BP event proxies at the Mavri Trypa coastal Peloponnesian Greece speleothem, sampling resolution 5 years (Finné et al., 2017), the Beirut speleothem at Jeita Cave, sampling resolution 7 years (Cheng et al., 2015), and the Iranian plateau Gol-e Zard speleothem, sampling resolution 2-15 years (Carolin et al., 2019).

What have we already learned about the Mawmluh Cave speleothems and the Indian Summer Monsoon?

Ön et al. (2019) similarly claim that the very high resolution Mawmluh Cave speleothems' 4.2 ka BP events (Berkelhammer et al 2013; Kathayat et al 2018) need be subject to falsifiability testing. That testing has already been done. The Mawmluh Cave speleothem proxies are the product not of the westerlies, but the Indian Summer Monsoon. The six published ISM proxy studies providing high resolution 4.2 ka BP megadrought events are Staubwasser et al., 2003, Dixit et al., 2014; Nakamura et al., 2016, Dixit et al., 2018; Giosan et al., 2018; and Giesche et al., 2019. The westerlies and the ISM were synchronously and abruptly diminished, displaced, and/or diverted at 4.2 ka BP. So, too, of course, were the East Asian, Indonesian-Australian, African, North American and South American monsoonal systems, extending even to

Antarctica (Weiss 2016).

References

Berkelhammer, M., Sinha, A., Stott, L., Cheng, H., Pausata, F.S.R., Yoshimura, K.: An Abrupt Shift in the Indian Monsoon 4000 Years Ago. Climates, Landscapes, and Civilizations, Geophysical Monograph Series 198, American Geophysical Union.10.1029/2012GM001207, 2012.

Boyer, P., Roberts, N., and Baird, D.: Holocene environment and settlement on the Çarşamba alluvial fan, south-central Turkey: Integrating geoarchaeology and archaeological field survey, Geoarchaeology, 21, 675-698, 2006.

Bradley, R. and Bakke, J.: Is there evidence for a 4.2 ka BP event in the northern North Atlantic region?, Clim. Past Discuss., https://doi.org/10.5194/cp-2018-162, in review, 2019.

Carolin, Stacy A., Walker, Richard T., Day, Christopher C., Ersek, Vasile, Sloan, R. Alastair, Dee, Michael W., Talebian, Morteza, Henderson, Gideon M.: Precise timing of abrupt increase in dust activity in the Middle East coincident with 4.2 ka social change, Proc Natl Acad Sci USA, 116, 67-72, 10.1073/pnas.1808103115, 2019.

Cheng, H., Sinha, A., Verheyden, S., Nader, F. H., Li, X. L., Zhang, P. Z., Yin, J. J., Yi, L., Peng, Y. B., Rao, Z. G., Ning, Y. F., and Edwards, R. L.: The climate variability in northern Levant over the past 20,000 years, Geophys Res Lett, 42, 8641-8650, 2015.

Dean, J. R., Jones, M. D., Leng, M. J., Noble, S. R., Metcalfe, S. E., Sloane, H. J., Sahy, D., Eastwood, W. J., and Roberts, C. N.: Eastern Mediterranean hydroclimate over the late glacial and Holocene, reconstructed from the sediments of Nar lake, central Turkey, using stable isotopes and carbonate mineralogy, Quaternary Sci Rev, 124, 162-174, 2015.

Dixit, Y., Hodell, D. A., Giesche, A., Tandon, S. K., Gazquez, F., Saini, H. S., Skinner, L. C., Mujtaba, S. A. I., Pawar, V., Singh, R. N., and Petrie, C. A.: Intensified summer

monsoon and the urbanization of Indus Civilization in northwest India, Sci Rep, 8, 4225, 2018.

Dixit, Y., Hodell, D. A., and Petrie, C. A.: Abrupt weakening of the summer monsoon in northwest India 4100 yr ago, Geology, 42, 339-342, 2014.

Filikci, B., Eriş, K. K., Çağatay, N., Sabuncu, A., and Polonia, A.: Late glacial to Holocene water level and climate changes in the Gulf of Gemlik, Sea of Marmara: evidence from multi-proxy data, Geo-Mar Lett, 37, 501-513, 2017.

Finné, Martin, Holmgren, Karen, Shen, C. C., Hu, H. M., Boyd, M., and Stocker, S.:Late Bronze Age climate change and the destruction of the Mycenaean Palace of Nestor at Pylos, PLoS One, 12, e0189447, 2017.

Giesche, Alena, Staubwasser, Michael, Petrie, Cameron A., Hodell, David A.: Re-examining the 4.2 ka BP event in foraminifer isotope records from the Indus River delta in the Arabian Sea, Clim. Past, 15, 73-90, https://doi.org/10.5194/cp-15-73-2019, 2019.

Giosan, Liviu, Orsi, William D., Coolen, Marco, Wuchter, Cornelia, Dunlea, Anne G., Thirumalai, Kaustub, Munoz, Samuel E., Clift, Peter D., Donnelly, Jeffrey P., Galy, Valier, Fuller, Dorian Q.: Neoglacial Climate Anomalies and the Harappan 1 Metamorphosis: Clim. Past, 14, 1669-1686, https://doi.org/10.5194/cp-14-1669-2018, 2018

Göktürk, O. M., Fleitmann, D., Badertscher, S., Cheng, H., Edwards, R. L., Leuenberger, M., Fankhauser, A., Tüysüz, O., and Kramers, J.: Climate on the southern Black Sea coast during the Holocene: implications from the Sofular Cave record, Quaternary Sci Rev, 30, 2433-2445, 2011.

Göktürk. O.M.: Climate in the Eastern Mediterranean through the Holocene inferred from Turkish stalagmites. Inauguraldissertation, Universität Bern, 2011.

Hansson, Sven Ove: Falsificationism Falsified, Foundations of Science, 11, 275-286, 2006.

Kathayat, G, Cheng, H., Sinha, A., Berkelhammer, M., Zhang, H., Duan, P., Li, H., Li, X., Ning, Y., and Edwards, R. L.: Timing and Structure of the 4.2 ka BP Event in the Indian Summer Monsoon Domain from an Annually-Resolved Speleothem Record from Northeast India, Clim. Past, 14, 1869-1879, 2018.

Kuhn, Thomas: The Structure of Scientific Revolutions, Chicago, University of Chicago Press, 1962.

Kuzucuoglu, C., Dörfler, W., Kunesch, S., and Goupille, F.: Mid- to late-Holocene climate change in central Turkey: The Tecer Lake record, The Holocene, 21, 173-188, 2011.

Lakatos, Imre: The Methodology of Scientific Research Programmes (Philosophical Papers: Volume 1), J. Worrall and G. Currie, (Eds.), Cambridge: Cambridge University Press, 1978.

Lemcke, G. and Sturm, M.: $\delta$18O and Trace Element Measurements as Proxy for the Reconstruction of Climate Changes at Lake Van (Turkey): Preliminary Results. In: Third Millennium BC Climate Change and Old World Collapse., Dalfes H.N., K. G., Weiss H (Ed.), NATO ASI Series, 1: Global Environmental Change, Springer, Berlin, Heidelberg, 653-678, 1997.

Leng, M. J., Jones, M. D., Frogley, M. R., Eastwood, W. J., Kendrick, C. P., and Roberts, C. N.: Detrital carbonate influences on bulk oxygen and carbon isotope composition of lacustrine sediments from the Mediterranean, Global Planet Change, 71, 175-182, 2010.

Nakamura, A., Yokoyama, Y., Maemoku, H., Yagi, H., Okamura, M., Matsuoka, H., Miyake, N., Osada, T., Adhikari, D. P., Dangol, V., Ikehara, M., Miyairi, Y., and Matsuzaki, H.: Weak monsoon event at 4.2 ka recorded in sediment from Lake Rara, Himalayas, Quatern Int, 397, 349-359, 2016.

Ön, Z. B., Akçer-Ön, S., Özeren, M. S., Eriş, K. K., Greaves, A. M., and Çağatay, M.

N.: Climate proxies for the last 17.3 ka from Lake Hazar (Eastern Anatolia), extracted by independent component analysis of $\mu$-XRF data, Quatern Int, 486, 17-28, 2018.

Ön, Z. Bora, Greaves, Alan M., Akcer-Ön, Sena, Ozeren, M. Sinan: Comment on H. Weiss' review of "Is there evidence for a 4.2 ka B.P. event in the northern North Atlantic region?", Clim. Past Discuss., https://doi.org/10.5194/cp-2018-162-SC2, 2019.

Popper, Karl: Conjectures and refutations. The growth of scientific knowledge, New York, Basic Books, 1962.

Pustovoytov, K., Schmidt, K., Taubald, H.: Evidence for Holocene environmental changes in the northern Fertile Crescent provided by pedogenic carbonate coatings, Quaternary Res, 67, 315-327, 2007.

Roberts, N., Reed, J. M., Leng, M. J., KuzucuoÄ§lu, C., Fontugne, M., Bertaux, J., Woldring, H., Bottema, S., Black, S., Hunt, E., and KarabiyikoÄ§lu, M.: The tempo of Holocene climatic change in the eastern Mediterranean region: new high-resolution crater-lake sediment data from central Turkey, The Holocene, 11, 721-736, 2001.

Ülgen, U. B., Franz, S. O., Biltekin, D., Çagatay, M. N., Roeser, P. A., Doner, L., and Thein, J.: Climatic and environmental evolution of Lake Iznik (NW Turkey) over the last âĹij4700 years, Quatern Int, 274, 88-101, 2012.

Weiss, Harvey: Global megadrought, societal collapse and resilience at 4.2-3.9 ka BP across the Mediterranean and west Asia, PAGES Magazine 24, 62 - 63, 2016.

Weiss, Harvey: The 4.2 ka BP event in the northern North Atlantic. Clim. Past Discuss., https://doi.org/10.5194/cp-2018-162-RC2, 2019.

Please also note the supplement to this comment:
https://www.clim-past-discuss.net/cp-2018-162/cp-2018-162-RC3-supplement.pdf
* * *
**Anatolia 4.2 ka BP event paleoclimate proxy sites**

| Proxy site | Lat. | Long. | Publication | status | Resolution/yrs |
|---|---|---|---|---|---|
| Sofular Cave | 41.417 | 31.95 | Göktürk et al 2011 | wet | 5 |
| Gulf of Gemlik | 40.463 | 28.895 | Filikci et al 2017 | dry | >200 |
| Lake Iznik | 40.433 | 29.508 | Ülgen et al 2012 | dry | - |
| Lake Tecer | 39.431 | 37.084 | Kuzucuoğlu et al 2011 | dry | - |
| Eski Acigöl | 38.547 | 34.546 | Roberts et al 2001 | uncertain | 85 |
| Lake Hazar | 38.5 | 39.3 | Ön et al 2018 | wet | 175 |
| Nar Gölü | 38.34 | 35.456 | Dean et al 2015 | dry | 5 |
| Konya Lakes | 37.483 | 33.45 | Boyer et al 2006 | dry | - |
| Koçain Cave | 37.233 | 30.712 | Göktürk 2011 | cold | 2 |
| Göbekli Tepe | 37.223 | 38.923 | Pustovoytov et al 2007 | dry | >100 |
| Golhisar Gölü | 37.117 | 29.6 | Leng et al 2010 | dry | 98 |
| Lake Van | 38.592 | 42.847 | Lemcke, Sturm 1997 | dry | 116 |

**Fig. 1**                                                                       **Weiss 2019 CP**

**Fig. 1.** Weiss 2019 Apr 3 Fig 1 Anatolia 4.2 ka BP event paleoclimate proxy sites

[Figure]

**Fig. 2.** Weiss 2019 Apr 3 Fig 2 Lake Hazar del13C and del 18O

---

## Author Comment (AC2) · 23 May 2019

Please see Supplementary file for color version in which responses to reviewers are provided in red.

Our response to RC2 has already been provided on line, and we have nothing further to add. We see no reason to comment on SC2, which is really addressed to the reviewer RC2

SC1 The caption of Figure 2 says: Holocene August SSTs at various locations in the northern North Atlantic (Anderson et al., 2004) and alkenone-based SSTS from sediment cores along a N-S transect in the North Atlantic Current-Norwegian Current system. I might be wrong here, but the records seem to be all alkenone records, as

implied by the graph title.

Correct

I don't think there are any data from Andersen (not Anderson) et al., 2004, in the graph.

Correct!

Andersen et al., is a diatom paper. Yes, there is a diatom record from MD95-2011, but the data presented for this core looks like the alkenone data by Calvo et al., 2002.

Correct—there was a mistake in the caption, now corrected

Another thing, perhaps core locations could be added to Figure 1?

Done ——————————————————————— RC1 Summary and major suggestions This manuscript addresses an important and interesting central question – whether there is evidence for a prominent "4.2 ka event" in paleoceanographic and terrestrial paleoclimate records from the northern North Atlantic regions directly affected by the North Atlantic Current and East Greenland Current (excluding Iceland, which is covered in a parallel submission by other authors).

This paper can make a valuable contribution, but it will benefit from some major additions: It should be made clear exactly what records are considered in this study, and (if they are a subset of published records) why these records were chosen. For example, section 2 begins with "we consider a transect of sediment cores," but what sediment cores are referred to here, and why are these cores in particular singled out? Is the C1 "transect" a list of all available records the authors could find from the region? Or a subset of published records selected for some specific reason?

We examined almost all published sediment core records from the northern North Atlantic. It is quite clear that there are different signals recorded by alkenones and diatoms compared to forams. The former reflect near-surface conditions (in the photic zone or mixed later) thus providing paleo-SST estimates. in Figure 2, we plot alkenonebased SSTs so that the records are comparable in terms of the seasonality and depth represented; locations are shown in Figure 1. As far as we are aware, these are the only published alkenone-based SSTs from the region (cf. Leduc et al., 2010). We note that alkenone-based SSTs from sediment cores to the south and west of the area in Figure 1 show an almost linear decline in temperature from ~7ka B.P. to the present, but with no unusual anomaly around 4.2ka B.P. (see Sachs 2007, Figure 2B). It is thus fair to conclude that throughout the range of the Atlantic current, from (at least) ~48N to 75N, there is no significant anomaly around 4.2ka B.P. As noted on lines 89-90, Mg/Ca ratios and oxygen isotopes in forams, as well as foram assemblage changes show a different pattern, reflecting sub-surface conditions. Nevertheless, these proxies do not reveal an anomaly around 4.2ka either.

Choice of sites used in all parts of this study should be clearly explained (and justified if the study is assessing a subset of published records), preferably (for clarity) in a short "Methods" section (which is currently lacking from the paper).

This is now made clear for the paleoceanographic records discussed in Section 2 (on lines 56-60). For the terrestrial records it seems reasonably clear that we are reviewing the available lake sediment, ice core and glacier records from areas adjacent to the northern North Atlantic.

Site locations should be added to Figure 1. Optionally, a table listing all sites, locations, proxies, and original publications would be useful.

The data sources and locations are now listed in the caption to Figure 1, and color-coded to link to Figure 2

In my opinion, all of the datasets described here should be shown in the paper's figures. (As an example, Fig. 4 is a nice way to visualize a large number of glacier reconstructions; although it is unclear whether that figure is updated with records obtained since 2009.) It would also be useful to include a statistical summary/ analysis of the data, as an objective test for a 4.2 ka event.

This would require a standardized data screening protocol (to assess dating control, data quality and analytical uncertainties, sampling resolution etc) (e.g. McKay & Kaufman 2014, Sci. Data, 1, 140026). We did not approach the published records in that way as we found it relatively straitforward to address the question posed: was there an unusual event at 4.2ka B.P.? That is quite easily answered by inspection of the records, as Figures 2-4 clearly demonstrate. Papers published since 2009 are discussed.

Very few of the data used to support the paper's main conclusion are shown, and there is no particular methodology of site selection or analysis described. Thus, with the information provided, it is impossible for readers to evaluate or appreciate the evidence. I personally trust the authors' expert knowledge of records from the region – but that alone is not a strong enough basis for their conclusions to be published in COP.

As with any review of the literature, we have attempted to examine any and all published papers that pertain to the question posed in the title of our paper: Is there evidence for a 4.2ka B.P. event in the northern North Atlantic region? That was our methodology—to read the literature and draw conclusions from the published records. We do not think it is necessary to reproduce all of the records that have no evidence for the 4.2ka B.P.—that would be a mammoth task. We have cited a large number of papers that provide the basis for our conclusion that there is no compelling evidence for a "4.2ka B.P. event" in the northern North Atlantic region, except for a few records that we explicitly mention.

The goal of this study is to assess whether a 4.2 ka event is a coherent feature of Holocene climate in the study region. Yet most of the text actually summarizes multi-millennial climate trends through the Holocene, and only relatively short sections of text discuss/evaluate evidence for 4.2 ka events in various regions.

In the first part of section 2, for example, lines 55-84 summarize multi-millennial Holocene trends, then lines 85-94 assess higher-frequency events and conclude there is no 4.2 ka event. Same for lines 96-118 (which review multi-millennial trends) vs.

[Figure]

lines 119-124 (which evaluate evidence for/against a 4.2 ka event). Given the goal of this study, I would expect to see relatively more extensive discussion of high-frequency events, and the evidence for/against a prominent, coherent 4.2 ka event. If statistical analyses are added as suggested above, that will help flesh out the discussion.

We feel that it is important to document the primary paleoclimatic signal in the region, which is the multi-millennial decline in temperature over the course of the mid to late Holocene. That is the signal that dominates almost all of the records examined. Superimposed on that are multi-decadal to century scale variations (as noted for example, on lines 98 and 191). Such variations are to be expected and so the question that guided us in this review was: is there evidence for an exceptional event at 4.2ka B.P. that stands out, beyond the range of such variability? It is clear that this is not the case, except in the few instances that we have noted, but is important to note, as we have done, that there was an overall deterioration in climate that began prior to 4.2ka B.P. That is a low frequency change, not an event. It does not serve the goals of this paper to examine in great detail all of the multi-decadal to century scale anomalies that occur in virtually every record.

Minor comments What convention is used to subdivide early from middle from late Holocene? e.g., the Abstract refers to the period 8-6 ka BP as part of the early Holocene, though in many subdivisions this would be considered part of the middle Holocene.

Reference to the mid-Holocene has been removed from the Abstract and the final paragraph

Depending on the desired geographic scope of this analysis, additional terrestrial records that could be included from east Greenland include Levy et al 2013 QSR (Bregne ice cap) and Lowell et al. 2013 QSR (Istorvet ice cap). Neither is ideal to capture subtle climate changes âĹij4.2 ka, but these records may help make the authors' point about idiosyncratic glaciation thresholds for individual glaciers/catchments.

It is not clear why these papers should be cited; we take the point that Lowell et al. refer to a gradient in ELA so that different parts of the ice cap may respond differently to a change in ELA, but they only address changes in the ice edge over the last ~2000 years. Furthermore, the discrepancies in their study, pointed out by Miller et al. (2013), make it difficult to rely on their conclusions. A better reference for this point is Geirsdottir et al., 2019, which we now cite in relation to this. The Bregne Ice Cap study seems less relevant except that it reveals no evidence for an abrupt change in climate around 4.2ka B.P.

Line 19: It appears this paper is to be part of a special issue focused on the 4.2 ka event, so I understand why there is not an introduction to the broader (global/hemispheric) significance/question of the 4.2 ka event. Nonetheless, it would be helpful for future readers (who may read this paper on its own) if the Introduction began with at least a couple of sentences of background on the 4.2 ka event more broadly, rather than beginning by discussing Bond events, to clarify why testing for climate shifts at 4.2 ka (vs. shifts correlating with other Bond events) is of particular interest.

Text has been amended in the opening paragraph to set the stage for why this review is relevant.

Line 37: The Geirsdottir review of Icelandic records appears to be published (or nearly). These findings from Iceland seem highly relevant to the current study and should be briefly summarized and discussed somewhere (Section 3.3?).

The Icelandic evidence is now discussed in Section 3.2, with reference to the work of Geirsdottir et al (2019)

Line 75: "in all cases" SST cooling specifically? Meaning all published studies? Or cores in Fig. 2?

This referred back to the previous sentence where we say "in all proxies that are indicative of conditions in the photic zone". We removed the second phrase "in all cases" to clarify this, since we already stated it.

163: Omit extra comma after "at"

OK

Line 181: Why compare the coldest and warmest 20-year periods of the Vinther record, rather than stick to describing millennial-scale changes? Comparing the warmest and coldest millennia would seem more consistent with the rest of the discussion of multi-millennial scale changes. As currently written (e.g. followed immediately by the phrase "Superimposed on the long-term temperature decline. . .") 4.9 C comes across as an estimate of the long-term, multi-millennial temperature change inferred from the Vinther reconstruction (for which it would be a major overestimate).

OK, we amended this by calculating the difference between the most recent millennium and the period from 7-8ka b2k, and 8.5-9.5b2k. We now state [lines 188-190]: " The warmest (7-8ka b2k) and coldest millennia (0-1ka b2k) differ in temperature by ∼2.35°C (assuming no change in the seasonality of snowfall on the ice sheet)"

Line 249: The concluding sentence states that "Although a few records do show a distinct anomaly around 4.2 ka BP (associated with a glacial advance) . . . we interpret it as a local signal of overall climatic deterioration that characterized the late Holocene." Is this consistently the onset of (subsequently more-or-less continuous) glaciation, as opposed to a transient, short-term glacial advance? If the former, which is what I gather from the previous discussion, then stating that more clearly here would strengthen this final assertion of the paper.

We have amended the sentence to be even more explicit: "Over the last 5000 years, a series of multi-decadal to century scale fluctuations occurred, superimposed on an overall decline in temperature. Against this background of declining temperatures, three records in northwest Greenland and Ellesmere Island show an unusual warm

[Figure]

anomaly around 4.2ka B.P., and a few others (in SE Greenland, Iceland and western Svalbard) show a cold anomaly, associated with a glacial advance. We interpret these as local events – simply one glacial advance of many that occurred in response to the overall climatic deterioration that characterized the late Holocene."

Fig 1. Locations of study sites mentioned in the text should be shown in this figure. Fig. 2. Why are data plotted only from these sites?

Figure 1 now shows the location of sites used in Figure 2. The reason for the site selection is discussed in the related text.

Fig 3. It would be useful to see Kobashi et al. 2017 also plotted, given the different sub-millennial variability in that record.

Done—added in Figure 3d.

Please also note the supplement to this comment:
https://www.clim-past-discuss.net/cp-2018-162/cp-2018-162-AC2-supplement.pdf

---

## Author Comment (AC4) · 22 Jun 2019

In our response to RC 1 we had included the following statement, rather than explicitly responding to RC 3.: "We see no reason to comment on SC2, which is really addressed to the reviewer RC2" But in the response from Weiss to SC2, he repeats the same argument made in his earlier review, that we "did not...consider about 50 available and relevant high resolution proxies, most of which document abrupt 4.2 ka BP events in Svalbard, Norway,Sweden, Denmark, Faroe Islands, Ice-land, Greenland, and Ellesmere Island"

We have already addressed this complaint in our response to RC2 and have nothing further to add. We took care to review the relevant literature and do not consider that

we undertook, "skewed and truncated sampling".

The rest of the text of RC3 is an argument between him & others working on a site in Turkey, which is totally unrelated to our submission. This appears to be a rather too personal attack on the authors of that comment and highlights the fact that Weiss is all too eager to dismiss anything that does not comport with his zealous belief in a global "event" at 4.2ka B.P., rather than consider that some areas may not actually record such evidence. That is unfortunate because it is only by carefully assessing where there is strong evidence, and where there is not, that we will be able to understand what the cause of such an event might have been. And incidentally, no objective author should characterize a comment as jejune–that is in itself puerile and inappropriate for a civilized scientific discussion.

---

## Author Response (AR1)

We appreciate the perspective of the Editor and the fair handling of what turned out to be a rather contentious manuscript (to our surprise!). We have made final revisions in accordance with his suggestions.

Perhaps so, but it seems clear that readers who are attracted to a Special Issue on "*The 4.2ka B.P. event*" must expect to find research that demonstrates something unusual occurred ***at that time***. In some papers that I reviewed, authors were referring to a "4.2ka BP" event, but their anomalies were centuries later (& not because of an uncertain chronology). We did not focus on late Holocene climatic anomalies in general, but on a very specific "event" that originated with Harvey Weiss's work on the Akkadian collapse. To understand the broader significance of that "event", we need to be quite precise on the temporal and spatial dimensions of what happened. I hope the collection of papers will help shed light on that, but the assessments need to be objective. The goal is not to prove a pre-conceived idea.

At line 31 some citation are necessary (why not Weiss 2016 or similar?)
Citation to the review by Weiss (2017) has been added

Line 69: just a comment: from the showed records also the 8.2 vent is not always represented….
OK—fair point

Line 96-97 from the text alone it is unclear which is the paper related to this well-resolved and well-dated reccors (is Berner et al., 2011?) Please clarify.
Sorry—citation to Sejrup et al., 2011 has been added

3.2 Iceland. It was important to insert this part. But probably it should be necessary spend few more words somewhere on why here it is reasonable to think that some evidences are present.
OK—we have added this preamble to the text:
"*Iceland is in a central location to experience major changes in the major oceanic and atmospheric circulation patterns of the North Atlantic*". We also amended the last sentence, thus:
"*Of the two lakes in NE Iceland that did not have a tephra in the sediments, one (Skoravatn) shows an abrupt change at 4.2ka B.P., while the other (Tröllkonuvatn) does not, making it difficult to draw conclusions about the impact of the eruption on changes recorded at that time*"

In general one reader would be very happy if the authors (very short eventually) try to draw inferences about why in this area the 4.2 event is neither particularly visible nor particularly prominent. I totally agree with Weiss considering the fact this event is evident in other regions, and I also agree with the authors that the absence is some region can help in understanding better the origin of this event. A short conclusion on that is probably necessary.
OK. We have added this to the conclusions:
"*Given that the northern North Atlantic is a key region for the formation of deepwater, which has consequences for the overall global oceanic circulation (the "conveyor belt"), the absence of a strong signal of an abrupt climatic event at 4.2ka B.P. suggests that—whatever the cause of changes seen elsewhere-- it is unlikely that the North Atlantic Ocean circulation played a driving*

*role. If this conclusion is correct, it requires that the cause of the 4.2ka BP event be sought elsewhere, in terms of direct radiative forcing (possibly due to explosive volcanic events, or earth surface aerosols resulting from aridity or—[less likely]-- solar forcing). Currently, none of these possibilities provide a compelling argument. The alternative is that the observed changes were a consequence of internal climate system variability, perhaps modulated by the overall decline in summer radiation across the northern hemisphere due to orbital changes, which are generally considered as the cause of neoglaciation in the late Holocene, the onset of which roughly corresponds to the 4.2ka event as described by Weiss (2017)".*

Figure 1 is not particularly informative and the caption is rather poor. Please adds some information on the oceanic currents and in the caption information on the symbols (including references) is mandatory.

OK—done. The symbol colors were linked to those in Fig 2, but this was not very clear so we numbered them & added the main ocean currents.

Finally I agree with RC1 a figure showing some of the records discussed (selected by the authors is OK, but it is difficult to follow without any figure) in the text will improve the "pleasure" of the reading.

We think the revised Figure 1 now satisfies the request of RC1. A new Figure 5 identifies the location of terrestrial sites mentioned in the text and figures.

I recommend adding a figure 5 including some records quoted in the text, which the authors indicate as more significant. Possibly adding a record, which show the event. Like from Iceland?

See note above. We think a reader should now be able to follow the discussion of sites with a strong (or absent) signal, by reference to Figure 5.